# FILOsofer: A TEE-Shielded Model Partitioning Framework based on Fisher Information-Guided LoRA Obfuscation

## Abstract

On-device machine learning makes DNN models visible as a white-box to users, leaving them susceptible to stealing attacks. Trusted Execution Environments (TEEs) mitigate this risk by isolating model execution, but executing entire models within TEEs is inefficient and slow. To balance security and performance, TEE-Shielded DNN Partitioning (TSDP) executes privacy-insensitive parts on GPUs while confining privacy-critical components within TEEs.

This work demonstrates that existing TSDP approaches remain vulnerable under large query budgets (e.g., >500 queries) due to non-zero information leakage per query, enabling attackers to gradually construct accurate surrogate models. To address this, we propose FILOsofer (Fisher Information-Guided LoRA Obfuscation), which uses Fisher Information to perturb a small subset of key weights, rendering the exposed weights inaccurate and producing uniform outputs, thereby safeguarding the model even under unlimited queries. We then design a novel cross-layer LoRA to efficiently restore authorized-user performance, storing only LoRA parameters in the TEE to eliminate information leakage while minimizing the performance overhead. This lightweight design also allows seamless extension to LLMs. We evaluate FILOsofer in both experimental and real-world settings, achieving over 10× improvement in security and more than 50× reduction in computational overhead compared to prior TSDP solutions.

## 1 Introduction

On-device machine learning improves latency and privacy by processing data locally, but it also exposes models to new threats such as unauthorized access, model stealing attacks, and membership inference attacks (Zhu et al., 2021; Yan et al., 2020; Rakin et al., 2022; Mehnaz et al., 2022; Zhang et al., 2023). The model stealing attacks essentially aim to clone the victim model's functionality without authorized access to its original training data or parameters. Prior work (Zhang et al., 2024b; Yuan et al., 2024; Rakin et al., 2022) shows that white-box access to GPU-deployed models allows adversaries to efficiently steal models by replicating weights and parameters, achieving high accuracy with minimal computational cost (Orekondy et al., 2020; Juuti et al., 2019; Hanzlik et al., 2021).

To mitigate these security risks, researchers have explored *two* defense strategies: *(i) Cryptographic approaches:* Methods such as Multi-Party Computation (MPC) (Juvekar et al., 2018) and Homomorphic Encryption (HE) (Gilad-Bachrach et al., 2016; Kim et al., 2022) aim to safeguard both input data and model parameters through algorithmic guarantees. Despite their strong theoretical protection, these techniques remain impractical for mobile and IoT deployment due to excessive computational overhead and non-trivial accuracy degradation. *(ii) Hardware-based defenses:* By leveraging Trusted Execution Environments (TEEs) (Zhang et al., 2024b; Hu et al., 2023), these methods achieve substantially lower overhead than cryptographic techniques. However, executing entire DNNs within TEEs is generally infeasible, as their computational performance is $50\times$ lower than that of GPU-based rich execution environments (REEs).

To balance security and efficiency, recent work proposes **TEE-Shielded DNN Partitioning (TSDP)** (Zhang et al., 2024b; Mo et al., 2020; Sun et al., 2020), which protects privacy-sensitive

portions inside TEEs while offloading the rest to REEs. To realize this idea, prior work explores different partitioning strategies. Some shield layers are based on depth (shallow, deep, or intermediate layers) (Shen et al., 2022; Elgamal & Nahrstedt, 2020; Mo et al., 2020) while others focus on non-linear layers (Sun et al., 2020; Zhang et al., 2024b). In addition to layer-based partitioning, obfuscation techniques keep obfuscated or quantized weights within TEEs to protect model confidentiality (Zhou et al., 2023; Sun et al., 2024).

Despite these advances, existing TSDP methods still face a ***critical limitation***: even if some layers and weights are hidden within TEEs, the partitioned model running on GPUs remains highly accurate. This accuracy enables adversaries to bootstrap surrogate models with correct architectures and weights. The state-of-the-art approach, TEESlice (Zhang et al., 2024b), introduced a mitigation strategy; however, our study shows that even with TEESlice, large query budgets allow adversaries to reconstruct accurate surrogate models, since small amounts of per-query information leakage can accumulate over time, which represents an inherent weakness shared by all TSDP methods.

To address this limitation, we propose **FILOsofer** (Fisher Information-Guided LoRA Obfuscation), which is motivated by two core insights: first, selectively perturbing a small fraction of critical GPU-exposed weights can degrade backbone accuracy and reduce information leakage; second, task utility can be efficiently recovered for authorized users using a parameter-efficient, LoRA fine-tuning mechanism. Specifically, FILOsofer perturbs a tiny fraction of GPU-exposed weights guided by Fisher Information, to both make the GPU-exposed weights inaccurate and enforce output uniformity across inputs, thereby preventing attackers from extracting *any* useful information from model outputs. For authorized users, an adaptive, cross-layer LoRA branch within the TEE restores near-original model performance efficiently, avoiding the need to store or reload obfuscated weights during inference as previous obfuscation methods Zhou et al. (2023). User authorization is enforced via standard cryptographic protocols implemented using the TEE. A constraint-aware joint-training algorithm further optimizes the trade-off between minimal weight obfuscation and the smallest LoRA branch size, ensuring both secure and effective model deployment. The contributions of this paper are summarized as follows:

- We conduct a systematic evaluation of existing TSDP approaches and show that none of them can prevent information leakage, allowing attackers to incrementally reconstruct the model via model stealing attacks as query budgets increase.

- We propose FILOsofer, a novel TSDP framework that defends against model stealing attacks even under unlimited query budgets, while supporting low-latency inference on edge devices. FILOsofer combines Fisher-guided obfuscation with a lightweight cross-layer LoRA recovery branch, jointly preventing information leakage, preserving predictive accuracy, and incurring minimal overhead.

- We comprehensively evaluate FILOsofer on both experimental and real-world devices (Jetson Orin Nano), demonstrating a 10× improvement in security against model stealing attacks with 50× lower computational overhead. We further show that this lightweight design extends seamlessly to LLMs, and we propose two adaptive attacks to validate the robustness of our method.

## 2 BACKGROUND

**Trusted Execution Environments (TEEs)** TEEs offer strong confidentiality and integrity guarantees in untrusted environments by providing two key features: *execution isolation* and *code/data protection* (Costan & Devadas, 2016). Isolation is achieved through the physical separation of hardware and memory between protected and untrusted worlds. Code and data protection rely on cryptographic techniques such as encryption and message authentication codes (MACs). Both features depend on a distinct hardware/software runtime environment, known as the trusted computing base (TCB), which operates correctly even under a fully compromised operating system (OS).

**TEE-Shielded Secure Inference** To address the latency limitations of TEEs, TSDP refers to selectively protecting only parts of a DNN model within the TEE, instead of the entire model. This reduces inference latency and effectively converts white-box attacks into black-box ones. Table 1 outlines existing TEE-shield methods and their weaknesses. Consistent with the TEESlice setup, we evaluate the six representative baselines highlighted in the table.

Table 1: We categorize prior work relevant to TSDP and highlight the representative studies that are empirically evaluated in this paper.

| Category | Name | Venue | Methods | Weakness |
|---|---|---|---|---|
| Shallow Layers | Serdab Elgamal & Nahrstedt (2020) | CCGRID 2020 | Put the layers closer to the input inside the TEE | No protection on other layers and outputs |
| | Origami Narra et al. (2019) | Arxiv 2019 | | |
| Deep Layers | PPFLMo et al. (2021) | MobiSys 2021 | Put the layers closer to the output inside the TEE | No protection on other layers and outputs |
| | DarkneTZ Mo et al. (2020) | MobiSys2020 | | |
| | Shredder Mireshghallah et al. (2020) | ASPLOS 2020 | | |
| Intermediate Layers | AegisDNN Xiang et al. (2021) | RTSS 2021 | Choose intermediate layers inside the TEE | No protection on other layers and outputs |
| | SOTER Shen et al. (2022) | ATC 2022 | | |
| | TEESlice Zhang et al. (2024b) | S&P 2024 | | |
| Non-Linear Layers | ShadowNet Sun et al. (2023) | S&P 2023 | Put non-linear layers, such as activation layers, inside the TEE | No protection on other layers and outputs |
| | Magnitude Hou et al. (2021) | TDSC 2021 | | |
| | DarKnight Hashemi et al. (2021) | MICRO 2021 | | |
| | Slalom Tramer & Boneh | ICLR 2018 | | |
| Model Obfuscation | NNSplitter Zhou et al. (2023) | ICML 2023 | Perturbs critical weights and store them inside the TEE | Obfuscated weights need to be reload to REE for inference, no protection on the output. |
| | GroupCover Zhang et al. (2024a) | ICML 2024 | | |
| | TSQP Sun et al. (2024) | S&P 2025 | | |

**Low-Rank Adaptation (LoRA)** LoRA (Hu et al., 2022; Dettmers et al., 2023) is a parameter-efficient tuning method that adapts pre-trained models by injecting trainable low-rank matrices. Formally, consider a weight matrix $W_0 \in \mathbb{R}^{d \times k}$ in a neural network layer, where $d$ is the output dimension and $k$ is the input dimension. Traditional fine-tuning would update all parameters in $W_0$, resulting in $\mathcal{O}(dk)$ trainable parameters. In contrast, LoRA freezes $W_0$ and injects a learnable update in the form of a low-rank decomposition:

$$W = W_0 + \Delta W = W_0 + BA, \tag{1}$$

where $A \in \mathbb{R}^{r \times k}$, $B \in \mathbb{R}^{d \times r}$, and $r \ll \min(d, k)$ is the rank of the decomposition. The matrix $A$ projects the input into a lower-dimensional space of rank $r$ (the parameter tested in our experiment), and $B$ maps it back to the original output dimension. Only $A$ and $B$ are trained, reducing the number of trainable parameters from $\mathcal{O}(dk)$ to $\mathcal{O}(r(d + k))$, which is significantly smaller. Thus, LoRA achieves fine-tuning with minimal additional memory, compute, and storage cost, making it highly suitable for large-scale and resource-constrained scenarios.

## 3 THREAT MODEL

**Model Stealing.** We consider a deep neural network (DNN) deployed on resource-constrained edge devices equipped with Trusted Execution Environments (TEE). In this scenario, the attacker attempts to steal the victim model ($M_{\text{vic}}$) by exploiting access to its predictions and any unprotected components within the Rich Execution Environment (REE; e.g., GPU). Consistent with prior TSDP work and real-world deployments (Zhang et al., 2024b; Zhou et al., 2023; Zhang et al., 2024a), we assume that deployed models provide users with label-only outputs, an assumption further supported by a comprehensive survey of on-device ML systems (Sun et al., 2021).

**Adversary's Capabilities**. We consider the adversary's capabilities in three aspects. 1) The adversary first infers the protected model's architecture and weights from publicly available models ($M_{\text{pub}}$) in the REE, then initializes a surrogate model with these priors. 2) The attacker issues limited queries on carefully selected inputs and records the corresponding outputs to approximate the victim model's behavior. 3) The collected input–output pairs are then used to train the surrogate model. However, the portion of training data available for constructing such queries is restricted to fewer than 5% of the original training set, and query budgets are also restricted based on previous settings (Zhang et al., 2024b; Zhou et al., 2023; Orekondy et al., 2019).

## 4 SYSTEMATIC STUDY AND INSIGHTS

In this section, we conduct a systematic analysis of the limitations inherent in existing TSDP methods against model stealing attacks. By critically examining their empirical performance, we highlight key vulnerabilities and main gaps.

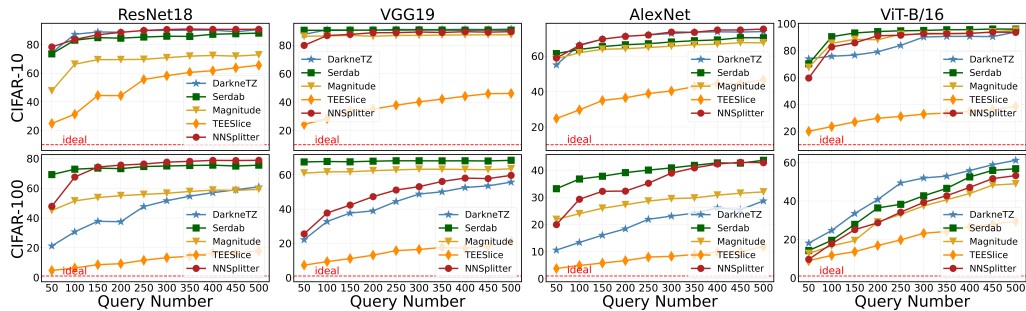

Figure 1: **Accuracy of surrogate model under varying query budgets**, with a dashed red baseline indicating the desired protection.

**Systematic Study:** To study this vulnerability, we select one representative method for each of the five categories of TSDP schemes in Table 1, and systematically evaluate model-stealing attacks across a range of query budgets. For the attack algorithm, we follow prior work (Zhang et al., 2024b; Sun et al., 2020) and adopt the KnockoffNet (Orekondy et al., 2019) as our representative modified model stealing attack. Figure 1 shows that all defenses fail against model stealing as queries increase. Even with modest budgets (e.g., 500 queries), surrogate accuracy rises sharply. For reference, the red dashed line marks an ideal baseline with consistently low accuracy.

The **fundamental weakness** of existing methods is that the partitioned model executed on GPUs remains accurate, enabling attackers to initialize surrogate models effectively. Since the model leaks mutual information between its weights and outputs, allowing attackers to gradually extract models as the number of queries increases. This is particularly concerning in edge environments, where attackers can perform effectively *unlimited* queries, highlighting the need for robust defenses. We summarize the key challenges and our solutions as follows:

**C1: Misleading attackers with inaccurate weights and useless outputs.** Excessive parameter modifications can degrade the model's predictive performance for legitimate users, while insufficient modifications may fail to prevent information leakage. **Solution:** Select the key weights and introduce tiny, targeted perturbations to guide the model's output toward a desired target label to decrease the GPU-exposed model accuracy and the leakage from outputs.

**C2: Retaining accuracy while minimizing TEE workload.** Storing and executing large portions of the model inside TEE introduces significant latency and resource consumption, which is impractical for edge devices. **Solution:** Unlike prior obfuscation methods (Zhou et al., 2023; Zhang et al., 2024a), we should avoid reloading weights to the GPU, preventing information leakage. LoRA provides an effective solution, and applying a single LoRA branch across multiple layers (cross-layer LoRA) can further enhance efficiency.

**C3: Reconciling obfuscation and recovery.** Obfuscation prevents information leakage, whereas recovery restores correct outputs for legitimate users; poorly designed recovery can weaken security or inadvertently leak sensitive information. **Solution:** Constraint-aware dynamic joint training, the obfuscation and recovery are jointly trained with attention to parameter sensitivity, enabling robust protection against attacks while effective recovery for authorized usage.

## 5 FILOSOFER

The overall system is shown in Fig 2. Our method integrates two components: Fisher-guided obfuscation, which perturbs key weights in critical layers to degrade backbone accuracy, and cross-layer LoRA, which restores task utility with adaptive rank updates. A constraint-aware joint-training algorithm balances these modules, ensuring obfuscation resists trivial recovery while LoRA maintains performance, thus achieving a trade-off between security and utility. For the online secure inference part, we deploy the cross-layer LoRA in the TEE and the obfuscated model in the REE, without reloading the obfuscated weights back to the REE.

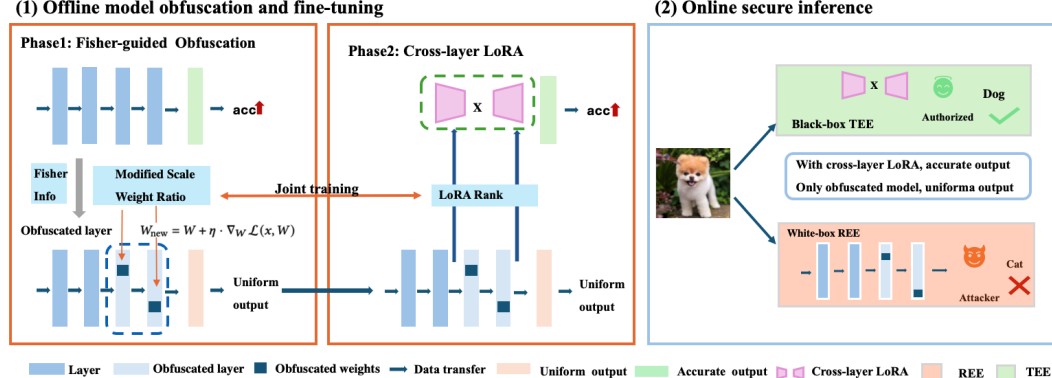

Figure 2: **Overview of FILOsofer**. The system combines Fisher-guided obfuscation, which perturbs critical weights to resist model stealing, with cross-layer LoRA, which restores utility via adaptive rank updates. During deployment, the obfuscated model runs in the REE, while the LoRA branch is executed inside the TEE for secure online inference.

## 5.1 FISHER INFORMATION GUIDED TARGET OBFUSCATION

The goal of obfuscation is to safeguard the edge-deployed model $M_{\text{vic}}$ against stealing by misleading adversaries with inaccurate weights and uninformative outputs, while keeping parameter alterations minimal to preserve usability for legitimate users. Formally, let $W$ denote the victim model's weights and $\Delta W$ a perturbation applied to a subset of them. The objective is to select $\Delta W$ that minimizes information leakage to the surrogate model $M_{\text{sur}}$ while ensuring recoverability:

$$\Delta W^* = \arg \min_{\Delta W \in \mathcal{R}} I\big(f(x; W); f(x; W + \Delta W)\big), \qquad (2)$$

where $I(\cdot; \cdot)$ denotes the mutual information over the input distribution $\mathcal{D}$ and $\mathcal{R}$ denotes the constraints of the perturbation (e.g., sparsity, magnitude). Let $f(x; W)$ and $f(x; W + \Delta W)$ denote the outputs of the original and obfuscated models. The mutual information is defined as

$$I(f(x; W); f(x; W + \Delta W)) = \sum_{y,z} p(y, z) \log \frac{p(y, z)}{p(y)p(z)}, \qquad (3)$$

where $y = f(x; W)$, $z = f(x; W + \Delta W)$, and $p(y, z)$ denotes the joint probability. If the perturbed output is independent of the input, i.e., $f(x; W + \Delta W)$ is constant for all $x \in \mathcal{D}$, then $p(y, z) = p(y)p(z)$. Substituting this into the mutual information formula yields

$$I(f(x; W); f(x; W + \Delta W)) = \sum_{y,z} p(y)p(z) \log \frac{p(y)p(z)}{p(y)p(z)} = 0. \qquad (4)$$

This confirms that when the obfuscated model's output is input-independent, no information can be inferred from its outputs. Thus, the problem becomes: identify the minimal perturbation that makes the obfuscated model's output input-independent (e.g., always output the same label).

**Fisher Information**, $F$, has been widely applied to evaluate the importance of parameters (Rissanen, 1996). Specifically, given a model $M$ with input $X$ and parameters $\theta_T$, the Fisher information can be calculated as:

$$F = \mathbb{E}\left[-\frac{\partial^2}{\partial \theta_T^2} \mathcal{L}(X|\theta_T)\right], \qquad (5)$$

where $\mathcal{L}$ is the loss function for the model. Intuitively, Fisher Information measures how sensitively the model output responds to small changes in its parameters; the more sensitive it is, the higher the Fisher Information. To enforce input-independent outputs, we perturb the weights that most strongly drive the model's predictions toward the target label $L_t$. Given input $x$ and target $L_t$, the Fisher information can be calculated as:

$$F_{L_t} = \mathbb{E}\left[\left(\frac{\partial \mathcal{L}(x, W)}{\partial W}\right)^2 \bigg| y = L_t\right]. \qquad (6)$$

We quantify perturbation intensity using the modification ratio $r$, defined as the fraction of weights obfuscated out of the total. To steer the model toward consistently predicting $L_t$, we perturb the key weights along the gradient of the target loss:

$$W \leftarrow W + \eta \cdot \nabla_W \mathcal{L}(x, W), \tag{7}$$

where $\eta$ is the scale factor. Experiments show that even small $\eta$ (e.g., $\leq 1e^{-4}$) are sufficient to reliably bias the model toward the target label. The complete algorithm is in Appendix 1. The full proof is provided in Appendix 9.10.

## 5.2 CROSS-LAYER LORA-BASED RECOVERY

While input-independent outputs effectively prevent information leakage, they degrade usability. Thus, it is essential to recover the model for legitimate users. LoRA fine-tuning introduces low-rank update matrices to the pre-trained weights, enabling efficient task-specific adaptation while keeping the obfuscated model weights **frozen**. We propose a **cross-layer LoRA** scheme to reduce recovery latency. Instead of attaching per-layer LoRA modules, we define a single branch $(A, B)$ spanning layers $\ell_s, \ldots, L$, where the entry layer $\ell_s$ is constrained to the last five layers and selected via Fisher information:

$$\ell_s = \arg\max_\ell \; \mathbb{E}\left[ -\frac{\partial^2}{\partial(W^{(\ell)})^2} \mathcal{L}(X|\theta) \right]. \tag{8}$$

Layers $\ell \geq \ell_s$ are obfuscated. During inference, the obfuscated backbone is computed in the REE, producing both the entry-layer activation $Z^{(\ell_s)}$ and a preliminary output $\tilde{y} = f_{\text{REE}}(X; W')$. Crucially, $\tilde{y}$ represents the degraded, inaccurate prediction derived from the perturbed weights $W'$. The TEE then receives $Z^{(\ell_s)}$ and applies the secure cross-layer LoRA parameters $(A, B)$ to synthesize the final prediction:

$$\hat{y} = f_{\text{TEE}}(Z^{(\ell_s)}; A, B) + \tilde{y}. \tag{9}$$

In this formulation, the term $f_{\text{TEE}}(Z^{(\ell_s)}; A, B)$ functions as a low-rank, task-specific residual learner. Mathematically, it is trained to predict the precise error vector required to compensate for the deviation introduced by the backbone obfuscation. By superimposing this secure corrective vector onto the erroneous preliminary result $\tilde{y}$, the system successfully reconstructs the accurate label $\hat{y}$ strictly within the trusted environment. This architectural decoupling effectively separates the model's utility from its bulk parameters: the REE executes the heavy but obfuscated computation, while the TEE handles the lightweight but critical recovery logic. Consequently, this design prevents the leakage of functional weights to the untrusted domain without incurring the high latency of full-model TEE execution, ensuring both robust security and computational efficiency.

## 5.3 OBFUSCATION AND RECOVERY TRADE-OFF

Obfuscation degrades backbone accuracy for security, while LoRA fine-tuning restores utility, creating a trade-off: excessive distortion hinders recovery and increases adaptation cost. To address this trade-off, we propose Constraint-Aware Obfuscation under Resource-Limited Adaptation (Details in Appendix, Algorithm 2). The algorithm iteratively maximizes obfuscation on the most sensitive layers while applying a resource-constrained cross-layer LoRA branch (e.g., limited in rank or parameter budget) to restore task performance. A rollback mechanism ensures that the LoRA-recovered accuracy never falls below a predefined threshold, guaranteeing recoverability. This procedure provides a realistic framework for maximizing model obfuscation under practical adaptation limits, particularly in edge device deployments, where recovery modules are inherently resource-constrained, and highlights the security–utility trade-off that arises in such constrained environments.

## 6 EXPERIMENTS

**Configuration.** Following the methodology outlined in TEESlice (Zhang et al., 2024b), we evaluate feasible configurations for the benchmarks introduced in Section 2. Specifically, for DarkneTZ (Mo et al., 2020) and Serdab (Elgamal & Nahrstedt, 2020), we vary the number of consecutive layers and report the results for the last three layers and first three layers, respectively. For Magnitude (Hou et al., 2021), we test configuration parameter 'mag_ratio' among $\{0.01, 0.1, 0.3, 0.5, 0.7, 0.9, 1\}$, where 0.01 is the recommended setting. For TEESlice (Zhang et al., 2024b), NNSplitter (Zhou et al., 2023) and GroupCover (Zhang et al., 2024a), we adopt the default configuration.

Table 2: **The accuracy of the surrogate model.** Green and Red boxes highlight the lowest and highest accuracy, respectively.

| | | No-Shield | DarkneTZ | | Serdab | | Magnitude | | NNSplitter | | TEESlice | | GroupCover | | Ours | Blackbox | |
|---|---|---|---|---|---|---|---|---|---|---|---|---|---|---|---|---|---|
| Budgets | Dataset | / | 50 | 5000 | 50 | 5000 | 50 | 5000 | 50 | 5000 | 50 | 5000 | 50 | 5000 | 5000 | 50 | 5000 |
| AlexNet | C10 | 81.58 | 66.01 | 80.97 | 74.08 | 80.26 | 68.42 | 78.08 | 58.9 | 75.26 | 24.79 | 66.51 | 13.60 | 47.70 | 10.00 | 19.37 | 74.20 |
| | C100 | 55.97 | 13.60 | 52.19 | 42.59 | 54.25 | 30.64 | 50.40 | 19.92 | 42.76 | 3.98 | 35.20 | 1.00 | 1.00 | 1.00 | 2.89 | 30.75 |
| | Image200 | 47.70 | 4.12 | 33.16 | 31.29 | 34.44 | 14.92 | 35.83 | 3.67 | 29.73 | 0.65 | 13.39 | 0.50 | 0.50 | 0.50 | 0.66 | 13.46 |
| ResNet18 | C10 | 93.07 | 86.84 | 92.79 | 86.86 | 92.25 | 73.26 | 88.63 | 78.30 | 90.66 | 23.87 | 65.21 | 10.00 | 57.60 | 10.00 | 23.12 | 65.92 |
| | C100 | 81.5 | 26.36 | 79.23 | 77.93 | 80.79 | 59.55 | 77.02 | 47.86 | 78.78 | 5.31 | 58.01 | 1.00 | 19.70 | 1.00 | 3.25 | 31.53 |
| | Image200 | 65.68 | 61.22 | 63.50 | 5.96 | 59.10 | 43.08 | 58.39 | 25.24 | 42.12 | 2.27 | 48.08 | 0.50 | 6.30 | 0.50 | 1.16 | 34.06 |
| VGG19 | C10 | 91.42 | 89.34 | 91.45 | 91.44 | 91.42 | 83.93 | 90.15 | 79.87 | 89.51 | 40.52 | 89.62 | 10.00 | 11.70 | 10.00 | 40.61 | 81.05 |
| | C100 | 70.39 | 22.63 | 67.71 | 69.07 | 69.85 | 59.95 | 65.34 | 25.50 | 59.62 | 7.31 | 48.97 | 1.00 | 1.30 | 1.00 | 7.09 | 49.34 |
| | Image200 | 63.23 | 60.89 | 61.17 | 5.24 | 52.66 | 24.67 | 45.20 | 13.28 | 49.26 | 2.83 | 43.72 | 0.50 | 2.90 | 0.50 | 2.54 | 42.09 |
| ViT-B16 | C10 | 97.69 | 67.96 | 97.12 | 65.54 | 94.99 | 95.26 | 97.92 | 59.55 | 93.56 | 22.64 | 97.64 | 12.60 | 39.70 | 10.00 | 20.63 | 95.02 |
| | C100 | 86.58 | 24.38 | 78.17 | 14.70 | 80.84 | 15.64 | 85.48 | 9.63 | 53.17 | 12.69 | 86.89 | 1.60 | 11.50 | 1.00 | 11.62 | 84.36 |
| | Image200 | 81.99 | 12.94 | 78.32 | 9.02 | 80.10 | 72.62 | 85.91 | 14.68 | 82.74 | 10.18 | 81.71 | 0.80 | 4.30 | 0.50 | 8.82 | 80.50 |

**Utility Cost Metric.** To evaluate the efficiency implications of different TSDP configurations, we adopt FLOPs as the primary utility cost metric. Following the setting proposed by TEESlice (Zhang et al., 2024b), %FLOPs is defined as the proportion of total floating-point operations (FLOPs) executed within the TEE, relative to the overall FLOPs of the full DNN model.

## 6.1 SECURITY GUARANTEE AND UTILITY COST

**Defense against Model Stealing** Table 2 presents the results of model stealing on four model architectures across three datasets under two attack budgets (50 and 5000 queries). The 'No-Shield' column denotes the baseline without any defense, reflecting that the surrogate model can directly copy the victim model. The 'Black-Box' setting assumes the attacker has no access to the model's weights and architecture but can use the input-output pairs to train the surrogate model.

Compared to existing defenses, strategies such as simple layer shielding or magnitude-based perturbations yield limited effectiveness. Similarly, TEESlice offers only moderate protection; while it modifies the model architecture, it critically leaves the model outputs unprotected, leading to potential leakage. In terms of other defenses, GroupCover demonstrates competitive performance by leveraging randomization strategies and mutual covering obfuscation. However, it fails to explicitly account for the mutual information leakage between the model parameters and the output. Consequently, although GroupCover performs well in many scenarios, its protection stability cannot be theoretically guaranteed. In contrast, our proposed method consistently achieves superior protection across diverse datasets and architectures. As illustrated in table 2, the accuracy of the surrogate model against our defense aligns strictly with the ideal random-guessing baselines (e.g., 10%, 1%, and 0.5%), demonstrating that our approach effectively eliminates information leakage.

Overall, our method maintains utility for authorized users while offering significantly stronger protection for unauthorized users by outputting a constant label. In addition, our framework supports a pay-per-query mechanism that can limit the number of model queries, ensuring long-term protection even under black-box access. Note that none of the prior TSDP-based methods *(a)* can distinguish between authorized and unauthorized users, and *(b)* can enforce query limits at the user level. This is the key difference between this work and prior art. There was no consideration given to the distinction between authorized and unauthorized actions in previous work, and we are the first to address this issue. We provide additional results and analysis for authorized vs. unauthorized user access in Appendix 9.7.

**Cross-Layer LoRA-based Recovery** We also test cross-layer LoRA recovery among different LoRA ranks. As shown in Table 3, higher LoRA ranks enhance recovery, yet even low ranks (e.g., rank 2) nearly restore original accuracy. Recovery scales differently across settings: shallow models (e.g., AlexNet on CIFAR-10) benefit from increasing LoRA rank, while deeper models or harder datasets (e.g., VGG19 on ImageNet200) demand higher ranks for comparable gains. In contrast, ViT-B/16 shows strong robustness and efficient recovery across datasets, with higher ranks even surpassing original accuracy, suggesting LoRA provides both restoration and performance gains.

**Efficiency** We further compare the computational efficiency and latency of our proposed method with existing approaches under the query size 500. Following the definition of the Utility Cost Merit outlined in Section 6, we estimate the number of floating-point operations (FLOPs) required for each method. Following prior work (Zhang et al., 2024b), we define Utility($C$) as the fraction of FLOPs

Table 3: **The recovery accuracy of cross-layer LoRA among varying LoRA rank**.

| Model | Dataset | Original | Obfuscated | LRank 2 | LRank 4 | LRank 8 | LRank 16 | LRank 32 |
|-------|---------|----------|------------|---------|---------|---------|----------|----------|
| AlexNet | C10 | 81.85 | 10.00 | 80.68 | 80.77 | 80.95 | 81.32 | 81.68 |
| | C100 | 55.97 | 1.00 | 55.62 | 55.93 | 55.95 | 56.30 | 56.92 |
| | ImageNet200 | 47.70 | 0.50 | 47.05 | 47.89 | 48.08 | 48.96 | 48.90 |
| ResNet18 | C10 | 93.07 | 10.00 | 92.56 | 92.92 | 92.98 | 93.04 | 93.42 |
| | C100 | 81.50 | 1.00 | 80.22 | 80.57 | 80.63 | 80.50 | 80.92 |
| | ImageNet200 | 65.68 | 0.50 | 49.22 | 50.03 | 52.13 | 53.82 | 55.20 |
| VGG19 | C10 | 91.42 | 10.00 | 90.76 | 90.85 | 90.94 | 91.48 | 91.65 |
| | C100 | 70.39 | 1.00 | 68.88 | 69.01 | 69.14 | 69.67 | 70.03 |
| | ImageNet200 | 63.23 | 0.50 | 60.74 | 61.10 | 61.77 | 62.66 | 63.32 |
| ViT-B16 | C10 | 97.69 | 10.00 | 97.43 | 97.73 | 97.67 | 97.80 | 97.96 |
| | C100 | 87.58 | 1.00 | 87.90 | 88.14 | 86.36 | 87.96 | 88.50 |
| | ImageNet200 | 81.99 | 0.50 | 82.69 | 83.27 | 83.43 | 83.58 | 84.16 |

Table 4: **Utility (%FLOPs) of prior works and FILOsofer**. Lower values imply lower utility cost, with %FLOPs being 0% for the white-box baseline and 100% for the black-box baseline.

| | Resnet18 | | | VGG19 | | | Alexnet | | | ViT | | |
|---|---|---|---|---|---|---|---|---|---|---|---|---|
| | C10 | C100 | ImageNet | C10 | C100 | ImageNet | C10 | C100 | ImageNet | C10 | C100 | ImageNet |
| DarkneTZ | 100.00 | 100.00 | 72.16 | 98.85 | 100.00 | 80.70 | 100.00 | 100.00 | 83.23 | 91.07 | 91.07 | 75.13 |
| Serdab | 100.00 | 100.00 | 96.54 | 100.00 | 100.00 | 98.62 | 100.00 | 100.00 | 95.72 | 91.73 | 100 | 83.40 |
| Magnitude | 100.00 | 94.71 | 78.43 | 100.00 | 87.43 | 75.57 | 81.18 | 90.58 | 71.82 | 100.00 | 72.20 | 66.91 |
| TEESlice | 3.80 | 5.33 | 3.80 | 0.34 | 0.37 | 0.31 | 12.48 | 12.48 | 8.75 | 7.24 | 8.51 | 8.92 |
| Ours | 0.0027 | 0.0027 | 0.0013 | 0.0032 | 0.0032 | 0.0021 | 0.0013 | 0.0013 | 0.0017 | 0.0069 | 0.0069 | 0.0069 |

that must run inside the TEE to match the security of the black-box baseline. NNSplitter (Zhou et al., 2023) uses an RL controller to select layers, making the protection level and TEE cost hard to quantify, since the modified weight ratio is not directly tunable.

As shown in Table 4, our method consistently achieves the lowest utility cost across all tested models and datasets, significantly outperforming state-of-the-art TEE-based defenses. We also observe that utility cost increases with dataset complexity, especially for large datasets such as ImageNet. This demonstrates that storing layers without importance selection is inefficient, whereas our Fisher information–based selection and cross-layer recovery scale robustly without sacrificing security, making it practical for deployment on resource-constrained edge devices.

## 6.2 PERFORMANCE ON REAL-WORLD DEVICES

To evaluate the practical performance of our methods, we deploy them on a NVIDIA Jetson Orin Nano, a widely used edge AI platform featuring a 6-core ARM v8.2 CPU, an Ampere GPU with 32 Tensor Cores, and 8 GB of LPDDR4x RAM. In addition to AI acceleration, Jetson provides hardware-level security features by ARM TrustZone, which enables secure execution by isolating trusted operations on ARM Cortex-A CPUs. We leverage OP-TEE to run trusted applications (e.g., our LoRA branch) within TrustZone.

We choose one ImageNet image as the input. As shown in Table 5, the inference latency of the backbone models executed on Jetson varies significantly by architecture, ranging from 12.4 ms for AlexNet to 91.7 ms for ViT-B/16, reflecting the growing computational demand of more complex models. For end-to-end latency, our evaluation over 10 runs showed a variation between 13.8–16.7 ms for AlexNet and 93.2–96.1 ms for ViT-B/16. In contrast, the latency of the corresponding LoRA recovery branches deployed within TrustZone remains consistently low across all models, below 1 ms in every case. This demonstrates that our LoRA-based design imposes minimal runtime overhead while providing robust model recovery.

Table 5: **Inference latency (per image) on Jetson Orin Nano**. The first row reports the inference latency (ms) of the obfuscated models executed on the GPU, the second row shows the latency (ms) of the LoRA branch deployed within the ARM TrustZone.

| Model | AlexNet | ResNet18 | VGG19 | ViT-B/16 |
|-------|---------|----------|-------|----------|
| GPU | 12.4 | 22.1 | 48.9 | 91.7 |
| TrustZone | 0.84 | 0.86 | 0.85 | 0.87 |
| **Overhead** | 6.3% | 3.7% | 1.7% | .9% |

## 6.3 APPLICABILITY TO LLMS

We evaluate our method on large language models (LLMs) but did not apply the knockoff-net attack, given the absence of an established framework for model stealing in this setting. Nevertheless, TDSP methods remain valuable for LLMs, as they provide mechanisms for usage protection and authentication (details in 9.3).

For our experiments, we choose LLaMA 3.2-1B and $\eta = 1e^{-5}$, $r = 5e^{-5}$, LoRA rank 8. In our experimental setup for LLMs, we initially fine-tuned the classifier layer using standardized system prompts constructed as "Question:" followed by the input query and candidate options. Subsequently, we designated the final layer (Layer 15) as the target for obfuscation; perturbations were applied to both this layer and the classifier head to effectively degrade baseline accuracy, while the cross-layer LoRA branch was employed to restore task utility. Further experiments investigating the influence of different layer selections are detailed in Appendix 9.10.

Table 6 shows the performance of three NLP benchmarks: GLUE-MNLI (3-class), ARC-Easy (4-class), and SciQ (4-class). Across all datasets, obfuscation consistently degrades accuracy, confirming that the obfuscated model produces low-quality outputs. With the cross-layer LoRA branch, predictive performance is restored, closely matching the original model and demonstrating an effective balance between security and utility for large language models. We also evaluate the inference latency of LLMs on edge hardware using the NVIDIA Jetson platform, showing that our approach is highly efficient.

This approach is particularly valuable for LLM deployment in pay-per-service scenarios. In such settings, models are executed on white-box edge devices, and users are billed per inference query. By obfuscating the backbone LLM, we prevent unauthorized copying or model misuse, while the lightweight cross-layer LoRA branch allows authorized clients to efficiently recover performance.

Table 6: **Application to LLM**. The First three lines are the accuracy of Llama3.2-1b. The last line is per-token latency.

| Dataset | Original | Obfuscated | LoRA |
|---|---|---|---|
| GLUE-MNLI | 78.14% | 33.49% | 78.09% |
| Arc_easy | 65.53% | 25.04% | 63.72% |
| SciQ | 91.40% | 25.31% | 90.26% |
| Latency (ms) | 86.3 (GPU) | 86.3 (GPU) | 0.88 (TrustZone) |

## 7 Resilience to Adaptive Attack

Based on the previous work (Zhou et al., 2023; Zhang et al., 2024a), we consider a more powerful adversary who seeks to optimize the performance of obfuscated models by employing advanced techniques, including norm clipping (Yu et al., 2021) and FisherPatch.

**Norm Clipping**: Following NNSplitter (Zhou et al., 2023), norm clipping (Yu et al., 2021) can be adapted to the weight level, where the adversary constrains weight perturbations within a scaled range of the modified parameters. The clipping interval is computed by scaling the minimum and maximum of $W + \Delta W'$ with a factor $t \in [0, 1]$, thereby effectively compressing the range to suppress outliers: $w_i \leftarrow \text{clip}(w_i, t \cdot \min(W + \Delta W'), t \cdot \max(W + \Delta W'))$.

As shown in Table 7, norm clipping fails across all threshold values $t$: large $t$ fails to clip the modified weights, whereas small $t$ excessively clips weights, significantly degrading classification performance. In contrast, norm clipping improves accuracy for NNSplitter (Zhou et al., 2023), as its magnitude-based obfuscation does not target specific directions; the clipped weights naturally revert toward the original decision boundary, partially restoring performance.

The ineffectiveness of this defense can be attributed to two main causes: 1) **Sparse directional perturbations** are resilient to norm bounds. Since only a few weights are altered, most values remain close to the original $W$, preserving the obfuscation even after clipping, especially when the perturbation scale $\eta$ is small (e.g., $1 \times 10^{-4}$). 2) **Semantic bias** is directional rather than magnitude-based. Perturbations align with the decision boundary of the target class. Even if clipping reduces their magnitude, the directional effect in weight space remains, sustaining misclassification.

Table 7: Accuracy of obfuscated models before $\rightarrow$ after applying norm clipping with varying $t$ from 0.1 to 0.9. For different $t$, the clipping accuracy remains low, which means clipping fails to restore performance across datasets.

| Model | C10 | C100 | ImageNet200 |
|---|---|---|---|
| Alexnet | 10.0 $\rightarrow$ 10.0 | 1.0 $\rightarrow$ 1.0 | 0.5 $\rightarrow$ 0.5 |
| Resnet18 | 10.0 $\rightarrow$ 10.0 | 1.0 $\rightarrow$ 1.0 | 0.5 $\rightarrow$ 0.5 |
| VGG19 | 10.0 $\rightarrow$ 10.0 | 1.0 $\rightarrow$ 1.0 | 0.5 $\rightarrow$ 0.5 |
| Vit-B16 | 10.0 $\rightarrow$ 10.0 | 1.0 $\rightarrow$ 1.0 | 0.5 $\rightarrow$ 0.5 |

Table 8: **FisherPatch results on ViT-B/16, CIFAR100.** From left to right: 1) Surrogate model accuracy varying scale factor (obfuscation ratio $= 5 \times 10^{-5}$); 2) Cross-LoRA recovery accuracy varying scale factor (obfuscation ratio $= 5 \times 10^{-5}$); 3) Surrogate model accuracy varying weight ratio (scale factor$=0.1$); 4) Cross-LoRA recovery accuracy varying weight ratio (scale factor$=0.1$).

| Scale | Top-$k$ | Acc. | Scale | LRank | Acc. | Ratio | Top-$k$ | Acc. | Ratio | LRank | Acc. |
|---|---|---|---|---|---|---|---|---|---|---|---|
| | 1k | 87.97 | | 16 | 80.59 | | 1k | 3.17 | | 16 | 79.37 |
| 0.05 | 10k | 85.17 | 0.05 | 32 | 83.44 | 1e-5 | 10k | 69.67 | 1e-5 | 32 | 82.82 |
| | 50k | 83.97 | | 64 | 85.91 | | 50k | 84.26 | | 64 | 83.17 |
| | 1k | 2.40 | | 16 | 79.40 | | 1k | 2.40 | | 16 | 79.40 |
| 0.1 | 10k | 2.14 | 0.1 | 32 | 81.93 | 5e-5 | 10k | 2.14 | 5e-5 | 32 | 81.93 |
| | 50k | 1.95 | | 64 | 81.95 | | 50k | 1.95 | | 64 | 81.95 |
| | 1k | 1.30 | | 16 | 77.18 | | 1k | 1.01 | | 16 | 79.24 |
| 0.5 | 10k | 1.00 | 0.5 | 32 | 82.15 | 1e-4 | 10k | 1.00 | 1e-4 | 32 | 82.24 |
| | 50k | 1.00 | | 64 | 82.85 | | 50k | 1.00 | | 64 | 82.21 |

**FisherPatch**: We also propose a novel adaptive attack in which the adversary is assumed to be aware that Fisher Information is used for obfuscation, but remains unaware of which specific layers are targeted. Consequently, the adversary computes Fisher information over the entire model, ranks the parameters, and fine-tunes only the top-$k$ weights using 5% of the training set. We evaluate obfuscation hyperparameters (scale factor $\eta$ and modified-weight ratio $r$), the choice of $k$ (number of retrained parameters), and the cross-layer LoRA used for recovery, reporting both surrogate model and recovery accuracies to quantify attack success and defense robustness.

Based on Table 8, we observe a trade-off between utility and security. Minor obfuscation can be easily recovered by the adaptive attacker due to small weight perturbations. As obfuscation intensifies, the attacker's ability to recover the model progressively diminishes. In contrast, our recovery method remains robust: while heavier obfuscation requires a larger LoRA branch, recovery accuracy stabilizes once the branch reaches a sufficient size (e.g., 32).

## 8 CONCLUSION

We proposed FILOsofer, a TSDP framework that achieves robust protection against model stealing attacks, even when the adversary is granted an unlimited query budget. FILOsofer employs a Fisher-guided obfuscation strategy that minimally perturbs a critical subset of weights, effectively ensuring that the model outputs leak no information to attackers. For authorized use, FILOsofer integrates a compact, cross-layer LoRA-based branch within the TEE to restore the model's performance. Extensive evaluation on both experimental and real-world devices (Jetson Orin Nano) demonstrates that FILOsofer increases resistance to model stealing by 10× while reducing computational overhead by 50×. Moreover, this lightweight design extends seamlessly to LLMs, and we introduce two adaptive attacks to further validate the robustness of our method.

## REPRODUCIBILITY STATEMENT

To facilitate reproducibility, we have uploaded the full implementation of our method, including training scripts, evaluation code, and configuration files, to an anonymous repository.[1] The repository will be made publicly available on GitHub after the review process.

## ETHICS STATEMENT

This work uses only publicly available datasets and does not involve human subjects or any private or sensitive information. We strictly follow all licensing terms and usage guidelines associated with the datasets employed. Our experiments are conducted in a controlled research setting, ensuring that no confidential or personally identifiable data is exposed or utilized. The contributions of this study focus entirely on methodological improvements in existing TSDP methods, aiming to enhance security and efficiency in machine learning systems. Moreover, all code and evaluations are intended for academic and scientific purposes, promoting reproducibility and responsible research.

---

[1]The anonymous link is available at: `https://anonymous.4open.science/r/fisher_obfuscation_lora_modify/`

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

# 9 APPENDIX

## 9.1 THE USE OF LARGE LANGUAGE MODEL

We employed a large language model (GPT-5) to assist in proofreading and enhancing the clarity of the manuscript. Specifically, the LLM was utilized for grammar checking, sentence restructuring, and minor language polishing to improve readability and linguistic precision. All scientific content, including the formulation of hypotheses, experimental design, data analysis, and interpretation of results, was entirely executed, and verified by the authors.

## 9.2 ALGORITHM

**Fisher-guided Obfuscation** We summarize FILOsofer obfuscation mechanism in Algorithm 1. The process takes three steps: (1) computing the gradients with respect to the target class $t$, (2) estimating the Fisher information based on these gradients, and (3) updating the weights with the highest Fisher information.

---

**Algorithm 1** Model Obfuscation with Fisher Information

---

**Require:** Loss function $\mathcal{L}$, model parameters $W$, dataset $\mathcal{D}$, target class $L_t$, selection ratio $r$, scale factor $\eta$
**Ensure:** Updated (obfuscated) model parameters $W'$
 1: **for** each $x \in \mathcal{D}$ **do**
 2:     compute loss for target class: $\mathcal{L}(x, W)$
 3:     compute gradient: $g(x) \leftarrow \nabla_W \mathcal{L}(x, W)$
 4:     accumulate squared gradients for Fisher estimate: $F_W \mathrel{+}= g(x) \odot g(x)$
 5: **end for**
 6: normalize Fisher estimate: $F_W \leftarrow F_W / |\mathcal{D}|$
 7: **for** each parameter tensor/block $w$ in $W$ **do**
 8:     select top-$r$ fraction indices by $F_W$: $\mathcal{I} \leftarrow \mathrm{TopK}(F_W, r)$
 9:     compute perturbation on selected indices: $\Delta w_{\mathcal{I}} \leftarrow \eta \cdot g_{\mathcal{I}}$
10:     apply perturbation: $w' \leftarrow w + \Delta w$
11: **end for**
12: **return** $W'$

---

**Constraint-Aware Obfuscation under Resource-Limited Adaptation** Algorithm 2 implements a systematic procedure to maximize model obfuscation while respecting the resource constraints of the adaptation module. Joint training uses the same setup as the cross-layer LoRA finetuning stage. For the obfuscation component, we only perform a single forward pass to calculate the Fisher information of each weight, which introduces negligible overhead. The LoRA branch requires finetuning and therefore needs access to the corresponding training dataset (e.g., CIFAR-100). To maintain efficiency while preserving effectiveness, rather than attaching a separate LoRA module to every layer, we design a cross-layer LoRA branch that spans all obfuscated layers, significantly reducing both parameters and training cost.

Line 1–5: Layer Sensitivity Selection. We first compute the Fisher information for each layer to measure its sensitivity (Line 2). The entry layer $\ell_s$ is chosen as the most sensitive layer, and all subsequent layers $\ell \geq \ell_s$ are defined as target layers $L_t$ for obfuscation. This ensures that the perturbation focuses on layers critical to model performance.

Line 7–8: Accuracy Evaluation and Stopping Criterion. After recovery, the LoRA-recovered accuracy $Acc_L$ is evaluated. The iteration continues until $Acc_L$ falls below a predefined threshold, ensuring that obfuscation is maximized without exceeding the adaptation capacity.

Line 10–16: Iterative Obfuscation and Recovery. For each iteration, the obfuscation function $F_{\mathrm{obf}}$ is applied to target layers $L_t$, progressively increasing the perturbation magnitude via parameters $(r_{\mathrm{obf}}, \eta)$. The resource-constrained LoRA branch is then applied across $L_t$ to restore task utility under the given adaptation budget.

Core Insight. This constraint-driven loop reveals that, under limited adaptation resources, one can systematically explore the maximum obfuscation a model can tolerate. By decoupling obfuscation strength from adaptation capacity, the algorithm balances security (through progressive perturbation) and utility (through resource-limited recovery), providing a principled mechanism to probe the

---

**Algorithm 2** Constraint-Aware Obfuscation under Resource-Limited Adaptation

---

**Require:** Backbone model $M$, dataset $D$, weight obfuscation function $F_{\text{obf}}$, cross-layer LoRA branch $h_{\text{LoRA}}(A, B)$, iterations $T$, initial obfuscation ratio $r_{\text{obf}}$, scaling factor $\eta$, fixed LoRA rank $r_l$, LoRA accuracy threshold $Acc_{\text{LoRA}}$

**Ensure:** Obfuscated model $M_{\text{obf}}$ and LoRA branch $h_{\text{LoRA}}$

1: **for** each layer $\ell$ with $|\ell, \ldots, L| < 5$ **do**                    ▷ Layer selection via Fisher Information
2:       Compute $F_\ell = \mathbb{E}\left[ -\frac{\partial^2 \mathcal{L}(X|\theta)}{\partial (W^{(\ell)})^2} \right]$
3: **end for**
4: Select entry layer $\ell_s = \arg\max_\ell F_\ell$
5: Define target layers $L_t = \{\ell \geq \ell_s\}$
6: **for** iteration $t = 1$ to $T$ **do**
7:       **if** $Acc_L < Acc_{\text{LoRA}}$ **then**                    ▷ Rollback if threshold violated
8:           **return** $M_{\text{obf}}, h_{\text{LoRA}}$
9:       **else**
10:          $r_{\text{obf}} \leftarrow r_{\text{obf}} \cdot \beta, \eta \leftarrow \eta \cdot \beta$                    ▷ Increase obfuscation
11:          **for** each layer $\ell \in L_t$ **do**                    ▷ Weight Obfuscation
12:              $W^{(\ell)} \leftarrow F_{\text{obf}}(W^{(\ell)}; r_{\text{obf}}, \eta)$
13:          **end for**
14:          Apply $h_{\text{LoRA}}(A, B)$ with fixed rank $r_l$ across $L_t$                    ▷ LoRA Recovery
15:          Store $M_{obf}$ and $h_{LoRA}(A, B)$
16:          $Acc_L = M_{\text{obf+LoRA}}(D)$                    ▷ Evaluate LoRA-recovered accuracy
17:      **end if**
18: **end for**

---

security–utility trade-off. Compared with prior obfuscation methods such as TEESlice, NNSplitter and GroupCover, our computational cost is substantially lower and the overall process is more stable. TEESlice requires iterative slice pruning and repeatedly training the pruned model, while NNSplitter relies on reinforcement learning to identify layers and weights, often requiring many search rounds. GroupCover applies both randomization strategies and mutual covering obfuscation, and need to calculate the mask process and nonlinear parts in TEE. In contrast, our approach needs only one Fisher pass plus lightweight cross-layer LoRA finetuning, making it significantly more efficient.

## 9.3 AUTHORIZED ACCESS AND TEE IMPLEMENTATION

Before presenting additional results, we first explain how authorized access is achieved in our setup.

To ensure security, we consider a *provisioning* step, where a remote trusted gateway and TEE agree on a "token" and a "session key" ($uk$) Zhao et al. (2019). The assumption is that users negotiate with such a trusted gateway (which knows the license key), and once proper authentications are made, the trusted gateway provisions the new token and session key and shares them with the authenticated user Zhao et al. (2019).

The session key is generated by leveraging a symmetric *license* key, $k$, using established cryptographic algorithms Zhao et al. (2019). All communication after this point (including communication required for token generation) is cryptographically protected (integrity and confidentiality) by $uk$.

The token is created by also leveraging the license, and can be defined as: $user_{id}, credits, expiry \| HMAC_k(...)$. The "credits" and "expiry" are optional but can be set if this is a pay-per-inference service.

During the inference phase, the remote user can directly query the model by creating an ARM TrustZone SDK call (i.e., Secure Monitor Call, SMC) with the token. Note that all communications are encrypted and authenticated using $uk$. The trusted app (TA) then verifies the token and accepts the request if credits remain and are unexpired. The TA then performs the inference and returns an encrypted response. Under this model, an unauthorized user, whether local or remote, cannot successfully query the model, as they lack access to the session key and valid tokens. Furthermore, the untrusted operating system is unable to infer any information, since all communication between the user and the TEE is encrypted and protected for both confidentiality and integrity.

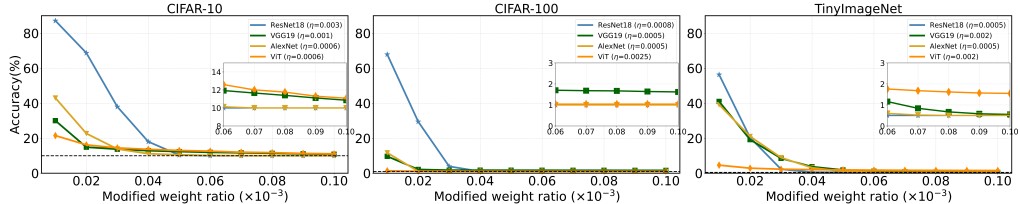

Figure 3: The influence of modified weight ratio with target label 3. For different models, different $\eta$ are applied. The inserted figure shows an amplified view of the x-axis in the range [0.06, 0.10].

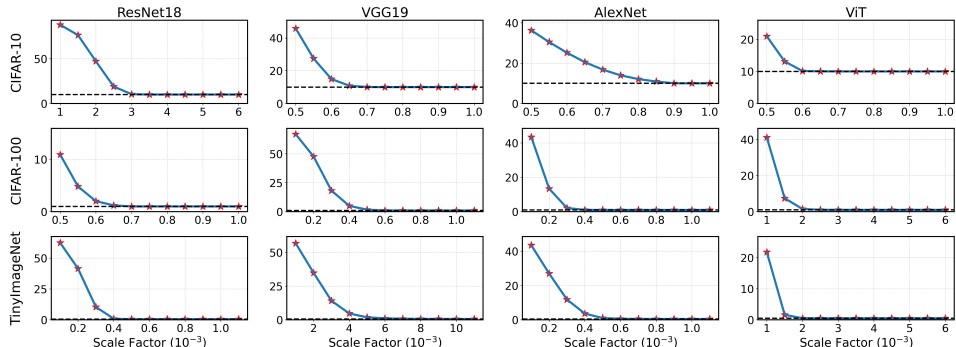

Figure 4: The influence of scale factor $\eta$ on different datasets and models.

## 9.4 EVALUATION ON MIA

We also evaluate membership inference attacks as downstream threats to demonstrate the effectiveness of our protection. Membership inference attacks (MIA) test whether an input sample $x$ belongs to the training dataset $D_{train}$. Formally, given query access to a target model $f_\theta$, the adversary constructs a hypothesis test between $H_0 : x \notin D_{train}$ and $H_1 : x \in D_{train}$, often leveraging prediction confidence or loss values.

Table 9: **Results of membership inference attack**.

| Dataset/Model | Serdab | DarkneTZ | Magnitude | NNsplitter | TEESlice | Ours | Blackbox |
|---|---|---|---|---|---|---|---|
| C10/ResNet18 | 66.06 | 65.13 | 59.28 | 50.00 | 50.00 | 50.00 | 50.00 |
| C10/VGG19 | 63.87 | 64.03 | 58.82 | 50.00 | 50.00 | 50.00 | 50.00 |
| C100/ResNet18 | 91.81 | 85.47 | 61.88 | 50.00 | 50.00 | 50.00 | 50.00 |
| C100/VGG19 | 87.80 | 84.66 | 71.48 | 50.00 | 50.00 | 50.00 | 50.00 |

The results show that obfuscated methods, like NNSplitter, is effective against membership inference attacks (MIA). By perturbing parameters, obfuscation reduces overfitting and diminishes the statistical gap between members and non-members in the output distribution $p_\theta(y|x)$, thereby weakening the adversary's likelihood test advantage.

## 9.5 IMPACT OF OBFUSCATION PARAMETERS

We further explore the factors that affect the effectiveness of model obfuscation. In particular, we examine the influence of the scaling factor $\eta$, weight modification ratio $r$, and target label $L_t$. Based on the experiment, we have the following findings:

**Scale Factor $\eta$.** Figure 4 shows the impact of the scale factor $\eta$ across different models (Resnet18, VGG19, AlexNet, and ViT) and datasets (CIFAR10, CIFAR100 and TinyImageNet200). As shown in the figure, different models and datasets favor different scaling factors; the optimal range for ResNet18 on CIFAR-10 is around $1e^{-3}$, while for ViT on the more complex TinyImageNet, it is in the much smaller range of approximately $5e^{-4}$. All effective values remain very small, a

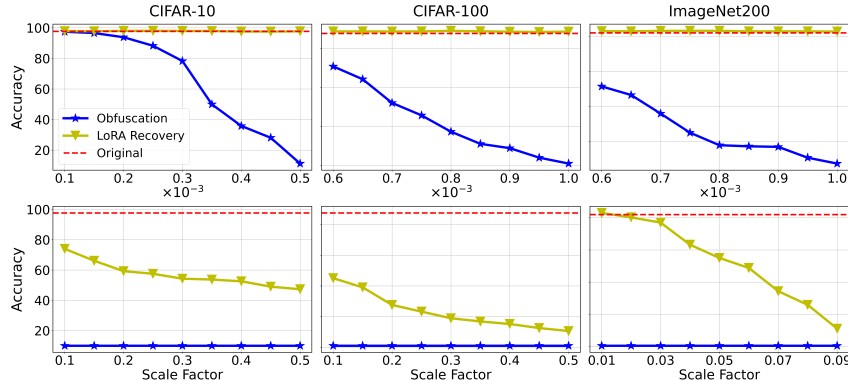

Figure 5: **The trand-off between Fisher-guided obfuscation and Cross-layer LoRA-based fine-tuning.**

characteristic that is advantageous for obfuscation as it makes the corresponding modifications to the input data difficult to detect.

**Weight Ratio** $r$. A small set of weights heavily influences predictions, but their identities vary across models and datasets. As shown in Figure 3, modifying key weights in ViT models reduces accuracy on CIFAR-10 and TinyImageNet, but the output does not consistently converge to the target label. When the scale factor is too small, even widespread perturbations fail to induce consistent misclassification. This underscores the importance of selecting an appropriate scale and confirms that only a small fraction of weights are truly critical to model behavior.

**Target Class**: Table 10 illustrates how obfuscation accuracy varies with different target labels, under the setting of $r = 0.00005$ and $\eta = 0.003$ for ResNet18, $\eta = 0.0007$ for VGG19, and $\eta = 0.001$ for AlexNet. By slightly increasing $\eta$, the accuracy drops to 10% (all output target labels) across all classes. Therefore, we carefully select $\eta$ to highlight the differences between target labels. The results show only minor differences across target labels, suggesting that the model's sensitivity is largely uniform regardless of the target.

Table 10: The evaluation of obfuscated target classes on CIFAR-10.

| Class | ResNet18 | VGG19 | AlexNet |
|-------|----------|-------|---------|
| 0 | 10.38 | 10.25 | 10.40 |
| 1 | 14.90 | 10.24 | 11.33 |
| 2 | 18.67 | 10.00 | 10.42 |
| 3 | 10.15 | 10.00 | 10.55 |
| 4 | 10.38 | 10.00 | 12.24 |
| 5 | 11.81 | 10.00 | 12.30 |
| 6 | 12.85 | 10.00 | 12.10 |
| 7 | 10.85 | 10.00 | 12.41 |
| 8 | 12.72 | 10.25 | 12.66 |
| 9 | 18.91 | 10.02 | 11.21 |

## 9.6 TRADE-OFFS BETWEEN FISHER INFORMATION OBFUSCATION AND CROSS-LAYER LORA-BASED FINE-TUNING

We also evaluate the trade-off between obfuscation and LoRA-based fine-tuning, which is shown in Fig 5. We set $r = 0.00005$, target label three, and LoRA rank two. By varying $\eta$, we control the degree of model obfuscation and observe the extent to which the LoRA branch fails to recover.

**Across Datasets.** The tolerance for obfuscation varies significantly by dataset complexity. On CIFAR-10, even aggressive perturbations (e.g., reducing accuracy to 10%) still allow LoRA to recover over 97% performance, showing strong robustness in simpler tasks. In contrast, CIFAR-100 and ImageNet200 exhibit much steeper trade-offs: small increases in obfuscation strength rapidly degrade recoverability, reflecting their higher label granularity and reliance on fine-grained features.

**Across Models.** ViT achieves better LoRA recovery than CNN-based models under the same obfuscation conditions, especially on CIFAR-100 and ImageNet200. This suggests that Transformer architectures offer more adaptable representations, even when key weights are perturbed.

**Joint Training Advantage.** These results validate the effectiveness of our joint training algorithm, which dynamically balances obfuscation strength and LoRA capacity.

Table 12: The accuracy of the whole model cross-layer LoRA fine-tuning with different LoRA ranks and fine-tuning data.

| | Percentage(%) | AlexNet | | | VGG19 | | | ResNet18 | | | ViT-B/16 | | |
|---|---|---|---|---|---|---|---|---|---|---|---|---|---|
| | LoRA Rank | 1 | 4 | 16 | 1 | 4 | 16 | 1 | 4 | 16 | 1 | 4 | 16 |
| C10 | 5% | 17.62 | 17.06 | 16.10 | 19.48 | 23.72 | 22.61 | 17.53 | 17.25 | 19.78 | 13.38 | 14.03 | 12.62 |
| | 10% | 17.22 | 17.61 | 16.53 | 20.45 | 23.44 | 22.78 | 17.76 | 18.59 | 20.93 | 13.63 | 14.73 | 13.28 |
| C100 | 5% | 11.47 | 12.95 | 10.45 | 12.20 | 12.85 | 12.84 | 11.05 | 11.83 | 12.03 | 22.49 | 15.62 | 13.67 |
| | 10% | 12.98 | 13.78 | 10.63 | 14.09 | 14.54 | 14.73 | 12.54 | 12.22 | 12.79 | 24.22 | 22.41 | 21.62 |
| ImageNet | 5% | 4.05 | 3.08 | 3.20 | 6.11 | 5.60 | 4.83 | 5.93 | 4.25 | 3.40 | 21.32 | 22.37 | 22.57 |
| | 10% | 8.78 | 3.83 | 3.25 | 6.36 | 5.69 | 5.76 | 9.39 | 6.42 | 5.24 | 21.49 | 23.64 | 29.36 |

## 9.7 ROBUSTNESS OF AUTHORIZED USER

In this section, we evaluate the robustness of our framework under a strict threat model: an authorized user who has legitimate access to the model's inference service and receives the correct, authorized labels. Unlike external adversaries who may only receive obfuscated outputs, an authorized user possesses the ground-truth input-label pairs.

Table 11 presents the accuracy of the surrogate models constructed by authorized users. The results indicate that access to correct labels is insufficient for successful model extraction when the underlying weights are obfuscated. As shown in the table, even with a budget of 5,000 queries and valid labels, the surrogate model accuracy remains exceptionally low (e.g., $< 18\%$ on CIFAR-10 and $\sim 1\%$ on CIFAR-100). This demonstrates that our weight obfuscation strategy effectively breaks the correlation between the observable weights and the correct functional behavior.

## 9.8 CROSS-LAYER LORA ADAPTIVE ATTACK DISCUSSION

We conducted a systematic evaluation of the impact of both the LoRA rank and the amount of available training data on the adaptive attack performance, results shown in Tab 12. Our findings suggest that these two factors exhibit a strong interdependence. Specifically, for a fixed LoRA rank, increasing the proportion of training data consistently leads to improved accuracy, as the model benefits from more representative and diverse training signals. For example, by increasing the training data from $5\%$ to $10\%$, ViT-B/16 accuracy becomes higher for all dataset.

However, the relationship between LoRA rank and performance is more nuanced. Contrary to the intuition that higher-rank adaptations might yield better results due to increased capacity, we observe that ex-

Table 11: Defense effectiveness against authorized users. The table reports the accuracy (%) of surrogate models trained by authorized users who have access to **correct labels**. Despite possessing valid input-label pairs, the adversary fails to achieve high accuracy due to the weight obfuscation.

| Model | Dataset | Surrogate Model Accuracy (%) | |
|---|---|---|---|
| | | 50 Queries | 5,000 Queries |
| VGG19 | CIFAR-10 | 10.00 | 17.25 |
| | CIFAR-100 | 1.00 | 1.00 |
| | ImageNet200 | 0.50 | 0.50 |
| ResNet18 | CIFAR-10 | 10.00 | 18.51 |
| | CIFAR-100 | 1.00 | 1.00 |
| | ImageNet200 | 0.50 | 0.50 |
| ViT-B/16 | CIFAR-10 | 10.31 | 14.87 |
| | CIFAR-100 | 1.41 | 6.83 |
| | ImageNet200 | 0.85 | 2.21 |

cessively high ranks can lead to suboptimal performance, particularly when the training data is limited. In such scenarios, large LoRA branches introduce a greater number of trainable parameters, which may not be adequately optimized given the data constraints. Also, larger parameter spaces lead to more complex loss surfaces, making training more sensitive to initialization and learning rates.

These findings reveal a fundamental challenge in the attacker's recovery strategy. Despite increasing the LoRA rank or leveraging a moderate amount of training data, the obfuscated base model imposes a structural bottleneck that restricts information flow. Consequently, even high-capacity LoRA branches struggle to compensate for the intentionally degraded base model, resulting in a

Table 13: Top-5 gradient-sensitive parameters per class and dataset. Each entry shows Layer[Index] of the most sensitive parameters.

| Label | CIFAR10 | CIFAR100 | TinyImageNet200 |
|-------|---------|----------|-----------------|
| **0** | layer4.1.bn1.bias[318]
layer4.1.bn1.bias[480]
layer2.0.bn2.bias[40]
layer2.0.downsample.1.bias[40]
layer4.1.bn1.bias[38] | layer4.1.bn1.bias[196]
fc.weight[17]
fc.weight[182]
layer4.1.bn1.weight[489]
fc.weight[310] | fc.weight[255]
fc.weight[280]
layer2.0.bn2.bias[55]
layer2.0.downsample.1.bias[55]
fc.weight[182] |
| **1** | layer4.1.bn1.bias[38]
layer4.1.bn1.bias[132]
layer4.1.bn1.weight[132]
layer4.1.bn1.bias[318]
layer4.1.bn1.weight[38] | layer2.0.bn2.bias[121]
layer2.0.downsample.1.bias[121]
layer1.1.bn1.bias[55]
layer2.1.bn2.bias[121]
fc.weight[529] | layer4.1.bn1.bias[461]
fc.weight[767]
fc.weight[792]
layer4.1.bn1.bias[259]
fc.weight[694] |

persistent gap from the original performance. This supports the robustness of the FILOsofer technique against adaptive fine-tuning attacks.

We observe that when knowing exactly which layers have been obfuscated, applying cross-layer LoRA directly to these layers enables effective recovery of the original model behavior (as demonstrated by the recovery methods summarized in Tab. 3). However, in the adaptive attack scenario, where the attacker knows nothing about the target layers and attaches a large LoRA branch only at the input and output of the model, recovery performance significantly degrades. This contrast reveals several key insights.

1) **Lack of Access to Obfuscated Semantics.** The obfuscation targets the last few layers of the model, where task-specific semantics reside. LoRA branches attached only at the input/output cannot directly influence or correct these corrupted internal representations, making recovery ineffective. 2) **Gradient Misalignment.** When fine-tuning is performed without targeting the actual obfuscated layers, the gradients flow through a corrupted backbone. This leads to poor alignment between the loss signal and the parameters that need adaptation, severely limiting learning efficiency. 3) **Input-Level Adaptation is Too Weak.** Adapting only at the input/output level essentially treats the backbone as a fixed black box. Without modifying the internal transformations, the model cannot recover class-separability or generalization, especially when its outputs are collapsed to a single label.

## 9.9 THE CHOICE OF OBFUSCATED WEIGHTS ANALYSIS

We present the top five most sensitive weights of ResNet18 across different datasets and target labels in Table 13. The results indicate that the specific sensitive weights vary significantly depending on both the model's training data and the chosen target class.

## 9.10 LLM LAYERS ANALYSIS

Table 14 presents the impact of layer-wise obfuscation on model performance using the SCIQ dataset. 'Both' refers obfuscate both attantion layer and mlp layer. With a baseline accuracy of 0.92, the experiments utilize a scale factor of 0.1 and a modified weight ratio of $10^{-4}$ to evaluate the sensitivity of different architectural components. The results reveal a significant disparity in robustness across layer depths and types. Specifically, the "Attention" and "Both" configurations demonstrate relative resilience in the initial layer (Layer 0), maintaining accuracies of 0.889 and 0.885, respectively. However, this robustness rapidly diminishes in subsequent layers, with accuracy dropping precipitously in the middle and later stages (e.g., reaching as low as 0.194 at Layer 9). In stark contrast, the MLP layers exhibit extreme sensitivity to gradient-based perturbations; accuracy collapses to approximately 0.24 across all layers immediately upon perturbation, regardless of layer depth. These findings empirically confirm that MLP modules are the primary bottleneck for adversarial robustness in this context, whereas attention mechanisms retain partial resilience in the earliest embedding stages.

Table 14: The LLM modified results.

| Layer | 0 | 1 | 2 | 3 | 4 | 5 | 6 | 7 | 8 | 9 | 10 | 11 | 12 | 13 | 14 | 15 |
|---|---|---|---|---|---|---|---|---|---|---|---|---|---|---|---|---|
| Both | 0.885 | 0.593 | 0.452 | 0.324 | 0.434 | 0.443 | 0.281 | 0.359 | 0.213 | 0.194 | 0.248 | 0.350 | 0.246 | 0.306 | 0.241 | 0.286 |
| Attention | 0.889 | 0.540 | 0.485 | 0.329 | 0.461 | 0.307 | 0.332 | 0.321 | 0.291 | 0.308 | 0.258 | 0.305 | 0.239 | 0.317 | 0.239 | 0.289 |
| MLP | 0.239 | 0.305 | 0.254 | 0.299 | 0.245 | 0.254 | 0.286 | 0.240 | 0.239 | 0.273 | 0.239 | 0.239 | 0.239 | 0.239 | 0.239 | 0.239 |

## 9.11 FUTURE DISCUSSION

**Scalability of large language models** A promising direction for future research is further exploring the applicability of FILOsofer to protect large language models (LLMs), which pose unique challenges beyond those addressed in our current work. First, the definition and evaluation of model stealing in the context of LLMs remain underexplored and ambiguous. Unlike classification models with clear prediction labels, LLMs operate in open-ended generation settings such as dialogue, summarization, or instruction following, making it difficult to measure what constitutes a successful attack. Second, our current approach is tailored to classification tasks and does not account for the nuanced and context-dependent outputs of LLMs. Obfuscating the model in such a way that it consistently degrades the utility of stolen outputs without harming legitimate usage requires more sophisticated techniques. Developing mechanisms that generalize to the diverse interaction modes of LLMs will be critical for securing them in real-world applications.

**Distributed deployment scenarios.** Another important direction for future exploration is the protection of models in distributed deployment scenarios, where a single model is partitioned and deployed across multiple edge devices. In such settings, different segments of the model are executed on separate devices, potentially increasing the system's vulnerability surface. Attackers may attempt to compromise a subset of devices to reconstruct the behavior of the partial model or launch collaborative attacks. Our current framework, FILOsofer, is designed under the assumption of a single-device deployment and does not yet consider inter-device communication or consistency under adversarial interference. Adapting FILOsofer to support secure distributed inference requires addressing challenges such as secure partition coordination, synchronization of obfuscation effects across devices, and minimizing communication overhead, all while maintaining strong security guarantees. Future work could explore integrating lightweight secure multi-party inference protocols or developing partition-aware obfuscation strategies tailored to distributed edge environments.

## 9.12 FISHER INFORMATION PROOF

**Notation and setup.** Let $x \sim \mathcal{D}$ denote inputs and consider a conditional model $p(y \mid x; W)$. We perturb parameters $W$ to $W + \Delta W$ with $\|\Delta W\|$ small. Denote the perturbed conditional output distribution by $p_{W+\Delta W}(z \mid x)$ and its marginal by $p_{W+\Delta W}(z) = \int p_{W+\Delta W}(z \mid x)p(x)\,dx$. We use $g_{L_t}(x) := \nabla_W \log p(L_t \mid x; W)$ and the target-class score $s(x; W) := \log p(L_t \mid x; W)$.

**Assumptions.**

1. The mapping $W \mapsto p(y \mid x; W)$ is twice continuously differentiable for each $x$.

2. The perturbation $\Delta W$ is sufficiently small so that Taylor expansions are valid and higher-order terms are negligible.

3. Score functions have bounded second moments and satisfy standard regularity conditions ensuring the interchange of expectation and differentiation (so that the Fisher information is well-defined).

**Lemma 1** (Mutual information identity)**.** *For any joint distribution $p(x, z)$,*

$$I(X; Z) = \mathbb{E}_{x \sim \mathcal{D}}\big[D_{\mathrm{KL}}\big(p(z \mid x) \,\big\|\, p(z)\big)\big].$$

**Lemma 2** (Local KL expansion; conditional Fisher)**.** *Under (A1)–(A3), for small $\Delta W$,*

$$D_{\mathrm{KL}}\big(p(\cdot \mid x; W) \,\big\|\, p(\cdot \mid x; W + \Delta W)\big) = \tfrac{1}{2}\,\Delta W^\top F(x; W)\,\Delta W + o(\|\Delta W\|^2),$$

*where*

$$F(x; W) := \mathbb{E}_{z \sim p(\cdot \mid x; W)}\big[\nabla_W \log p(z \mid x; W)\,\nabla_W \log p(z \mid x; W)^\top\big].$$

*Proof.* By Taylor expansion of the log-likelihood and using the score zero-mean property, the first-order term cancels and the leading term is quadratic in $\Delta W$; standard derivations in asymptotic statistics produce the displayed form. $\quad\square$

**Theorem 1** (Local quadratic approximation of mutual information). *Under (A1)–(A3), for sufficiently small $\Delta W$,*

$$I_{W+\Delta W}(X;Z) = I_W(X;Z) + \tfrac{1}{2}\,\mathbb{E}_{x\sim\mathcal{D}}\big[\Delta W^\top F(x;W)\Delta W\big] + o(\|\Delta W\|^2).$$

*Proof.* By Lemma 1,

$$I_{W+\Delta W}(X;Z) = \mathbb{E}_x\big[D_{\mathrm{KL}}(p(\cdot\mid x;W+\Delta W)\|p_{W+\Delta W}(\cdot))\big].$$

One can expand the integrand around $W$ taking into account that both the conditional $p(\cdot\mid x;W+\Delta W)$ and the marginal $p_{W+\Delta W}(\cdot)$ vary with $\Delta W$. Careful bookkeeping of first- and second-order terms, and using Lemma 2 for the conditional contribution, yields the stated quadratic term as the dominant second-order contribution. The remainder is $o(\|\Delta W\|^2)$. $\quad\square$

**Proposition 1** (Optimal local perturbation under a Fisher (KL) budget). *Define the population-averaged conditional Fisher $\overline{F} := \mathbb{E}_{x\sim\mathcal{D}}[F(x;W)]$. Consider the constrained problem (quadratic-budget approximation)*

$$\max_{\Delta W}\quad \mathbb{E}_x[g_{L_t}(x)]^\top \Delta W \qquad s.t.\quad \Delta W^\top \overline{F}\, \Delta W \le \varepsilon.$$

*If $\overline{F}$ is positive definite, the optimal direction is*

$$\Delta W^\star \propto \overline{F}^{-1}\,\mathbb{E}_x[g_{L_t}(x)].$$

*Proof.* This is a standard linear objective with quadratic constraint problem. The Lagrangian is $L(\Delta W,\lambda) = \mathbb{E}_x[g_{L_t}(x)]^\top \Delta W - \lambda(\Delta W^\top \overline{F}\Delta W - \varepsilon)$. Stationarity yields $\mathbb{E}_x[g_{L_t}(x)] = 2\lambda\overline{F}\Delta W$. For $\lambda > 0$ and invertible $\overline{F}$, the result follows. $\quad\square$

**Remarks.**

- The matrix $\overline{F}$ is the correct second-order (KL) metric for measuring the distributional change induced by $\Delta W$. Using an uncentered class-specific matrix $F_{L_t}^{(\mathrm{raw})} = \mathbb{E}[g_{L_t}g_{L_t}^\top]$ without centering is generally inconsistent with the KL expansion unless additional conditional assumptions are made.

- If one targets directly the sample-wise variance of the target score, then the proper quadratic cost is $\Delta W^\top \tilde{F}_{L_t}\Delta W$, where $\tilde{F}_{L_t} := \mathbb{E}_x[(g_{L_t}(x)-\bar{g})(g_{L_t}(x)-\bar{g})^\top]$ is the centered class-covariance and $\bar{g} = \mathbb{E}_x[g_{L_t}(x)]$.

**Feasibility and practical implementation.**

1. For a finite representative dataset $\{x_i\}_{i=1}^m$, enforcing $s(x_i;W+\Delta W) = c$ for all $i$ under the first-order model reduces to a linear system $G\Delta W = b$ with $G_{i,:} = g_{L_t}(x_i)^\top$. If $G$ has full row rank and the parameter dimension $p$ is large, a solution exists (minimum-norm solution $G^+b$).

2. For population-level exact independence $p_{W+\Delta W}(z\mid x) = p_{W+\Delta W}(z)$ for all $x$ is generically impossible with finite-dimensional $\Delta W$; thus one aims at minimizing distributional proxies (variance, mutual information, empirical KL) instead of exact equality.

3. In practice, $\overline{F}$ and $\mathbb{E}_x[g_{L_t}(x)]$ are replaced by empirical estimates and $F^{-1}$ by approximations.

**Targeted Fisher for Obfuscation**  To steer the model toward a target label $L_t$ and reduce input dependence, we define the gradient-based measure:

$$F_{L_t} = \mathbb{E}\Big[\big(\frac{\partial \mathcal{L}(x,W)}{\partial W}\big)^2\,\Big|\,y = L_t\Big]. \tag{10}$$

This is a non-standard, heuristic Fisher matrix that captures which weights most strongly influence the output toward $L_t$. Selecting the top weights according to $F_{L_t}$ ensures that perturbations are applied where they are most effective in controlling the output.

**Perturbation via Gradient Update**  The perturbation is applied along the gradient of the target loss:

$$W \leftarrow W + \eta \cdot \nabla_W \mathcal{L}(x, W), \tag{11}$$

where $\eta$ is a scale factor. By first-order Taylor expansion:

$$s(x; W + \Delta W) \approx s(x; W) + \nabla_W s(x; W)^\top \Delta W, \tag{12}$$

the perturbation increases the target class score while reducing output variance across inputs, approximately decreasing mutual information.

**Rationale and Limitations**  This strategy is justified based on three key points:

1. **Fisher-guided selection:** Perturbing weights with high $F_{L_t}$ effectively targets the most sensitive parameters that control the output, consistent with information-theoretic intuition.

2. **Gradient alignment:** Applying $\Delta W \propto \nabla_W \mathcal{L}(x, W)$ aligns the perturbation with the direction that maximally increases the target score, which locally reduces output variance across $x$.

3. **Approximate input-independence:** While exact input-independence cannot be guaranteed (because different $x$ have different gradients and the model is nonlinear), iterative or multi-sample perturbations can significantly reduce the output's sensitivity to inputs, decreasing mutual information in expectation.

Therefore, the perturbation strategy is theoretically justified as an *approximate mutual information minimization* scheme guided by Fisher information.

