# OpenReview forum: "FILOsofer: A TEE-Shielded Model Partitioning Framework Based on Fisher Information-Guided LoRA Obfuscation"
_ICLR.cc/2026/Conference — Submitted to ICLR 2026_

### Official Review · Reviewer_cuad · 2025-10-26

**Soundness:** 2
**Presentation:** 2
**Contribution:** 2
**Rating:** 2
**Confidence:** 4

**Summary:**

This paper proposes a new protection method that obfuscates model weights while protecting their low-rank components in a TEE. The approach defends against model-stealing attacks with lower inference overhead.

**Strengths:**

It leverages Fisher Information to quantify weight importance and compares against multiple baselines to demonstrate its superiority.

**Weaknesses:**

1. The selection of representative baselines lacks justification. In Table 1, Magnitude is chosen for non-linear layers, yet ShadowNet is a newer and more relevant method. For model obfuscation, NNSplitter is selected, but it has been widely shown to be insecure by GroupCover, and the more secure GroupCover is not included.

2. The contributions are unclear. The claim “We conduct a systematic evaluation of existing TSDP approaches” has already been done by TEESlice, so it is not a novel contribution. The paper should clarify how the evaluation differs from TEESlice and report GroupCover’s results under the same attacks.

3. Section 4 lacks implementation details. It is unclear whether the query numbers in Figure 1 and the “modest budgets (e.g., 500 queries)” apply per class or to the entire dataset. The settings for baseline methods—such as hyperparameters and TEE FLOPs—are not described. The attack results may vary depending on the hyperparameters. The meaning of the “ideal line” is also ambiguous, it should report black-box attack performance rather than 10% or 1%.

4. Regarding the method, it is unclear how the target label ( $L_t$ ) is selected—randomly or a chosen one? Protecting fewer than five layers implies that the remaining weights remain in plaintext, raising security concerns. There is no justification for how the LoRA-based recovery maintains accuracy. It is also unclear how output privacy is preserved—authorized users may still be adversarial, and the true label is returned outside the TEE. More details are needed on how attacks are performed against the proposed method.

5. In the evaluation, it is questionable why FILOsofer yields lower attack performance than black-box, since black-box is generally considered the most secure (TEESlice, under 5000 queries, performs nearly the same as black-box). The efficiency evaluation appears theoretical rather than based on real runtime measurements. In Table 5, only TrustZone time is shown; full end-to-end runtime—including TrustZone, GPU, and data transmission—should be compared.

Minor: The code repository link was expired during my review.

**Questions:**

Please see the weakness.

---

> ### Author Response · Authors · 2025-11-21
> **Answering the questions**
>
> We are very thankful for your comments and questions. Below we address the main concerns.
>
> 1. We thank the reviewer for the comment. Regarding ShadowNet, it has been evaluated in TEESlides. However, its performance is consistently poor: for example, on CIFAR-10 and CIFAR-100, ShadowNet achieves the worst accuracy on AlexNet, the second-worst accuracy on VGG19 for CIFAR-10, and the worst accuracy on VGG19 for CIFAR-100. Due to these consistently low results, we did not include it as a representative baseline.
> Regarding GroupCover, we agree that GroupCover is more recent and provides stronger security. We have now included GroupCover results in the updated version to provide a more comprehensive comparison. The results of GroupCover are very impressive, as they use sufficient randomization strategy and mutual covering obfuscation. However, **GroupCover does not explicitly account for the mutual information leakage between model and output, so the protection cannot be stably guaranteed**. As a result, while the method performs well in most cases, its security is not consistently ensured.
>
> 2. Thank you for this valuable feedback. We will include GroupCover’s results for completeness and direct comparison. Regarding our contributions, our systematic evaluation extends beyond TEESlice by specifically analyzing the impact of the number of queries on the robustness of TSDP defenses—**an aspect that TEESlice did not explore**. Our results show that, similar to other TSDP approaches, **TEESlice also degrades under large query budgets, revealing a previously unreported vulnerability**. We will clarify this distinction in the revised manuscript to better highlight how our evaluation and findings differ from prior work.
>
> 3.  For the queries setting, **we follow the same setting in TEESlice, ShadowNet, GroupCover, as the queries apply the entire dataset**. For the baseline setting, sorry for the confusion, we follow the same setting in Section 6 with both Configuration and Utility Cost Metric.
> As we mentioned in Section 4, black-box is not the ideal line as the input-output pair remains accurate, which can **leak mutual information between the model and the output**. Following the previous settings in TEESlice and GroupCover, the blackbox only protects the model weights but not the output. In GroupCover, we can also see that it performs better than black-box (e.g.,Resnet18, CIFAR100, the blackbox is 25.0 and groupcover is 18.6). We selected 10%, 1% as the ideal line because it represents the random accuracy which means the attacker gets no information about the model.
>
> 4.  We are sorry for the lack of clarity. **We protect fewer than five layers to maintain efficiency, and perturbing only these Fisher-critical layers is sufficient to disrupt the backbone’s predictive path**. After perturbation, the remaining plaintext layers no longer leak useful information, as the model’s outputs become nearly uniform. Accuracy is preserved because FILOsofer jointly trains a cross-layer LoRA branch inside the TEE, which compensates for the obfuscated layers. Since LoRA is parameter-efficient and inserted only across the perturbed layers, it can reconstruct the disrupted predictive path and recover nearly full accuracy (within <1% of the original).
> As for the true label, please note that we insert the LoRA branch to protect last few layers. That’s how the output is protected, because we use the lora branch across the last several obfuscated layers to recover accuracy and store them into TEE, so the true label is computed inside the TEE using the secure LoRA branch and released only to authenticated users through standard cryptographic protocols (TLS + attestation).
>
> 5.  While black-box access hides the model weights, **it does not alter the outputs, so the input-output pairs remain fully accurate**. As a result, black-box alone cannot prevent leakage of mutual information between the model and its outputs—adaptive attacks can still exploit these outputs to recover information. Consequently, although black-box is traditionally viewed as secure, FILOSofer achieves lower attack performance because it limits the exploitable information in the outputs themselves, not just the weights. This is consistent with prior observations: methods that do not perturb outputs (e.g., standard black-box) leak more information, while output-obfuscating strategies (e.g., GroupCover Resnet18, CIFAR100, the blackbox is 25.0 and groupcover is 18.6) provide stronger resistance even with fewer queries.

---

> ### Author Response · Authors · 2025-11-21
> **Answering the questions**
>
> 6. Thank you for the suggestion. You are right that reporting TrustZone execution in isolation does not show full end-to-end cost. Using our measured backbone latencies (Jetson) and measured TrustZone execution times (OP-TEE), **we measured the end-to-end latency as follows (numbers show the lower and upper-bound for 10 different runs)**: AlexNet: ≈13.8–16.7 ms; ViT-B/16: ≈93.2–96.1 ms. These results show that (i) the LoRA recovery executed in TrustZone contributes <1 ms of compute, and (ii) overall E2E runtime remains dominated by backbone inference (especially for large models such as ViT). We will add these E2E numbers to the revised manuscript (and also include the numbers for the LLM experiment) and, as recommended, include the exact measurement protocol (wall-clock timing, warm-up, number of runs, standard deviation) and the raw timing log in supplementary material.
>
> 7. Minor: Sorry for the inconvenience. Now the link provided in the paper is valid, you can check the code here: https://anonymous.4open.science/r/fisher_obfuscation_lora_modify/ . We will release the code in GitHub after the review process.

---

> > ### Comment · Reviewer_cuad · 2025-11-24
> > **Response to author rebuttal**
> >
> > Thanks for the author response. Some concerns have been addressed, but many remain unresolved.
> >
> > My main concern is the method itself. The authors repeatedly emphasize that the approach protects the output and that other baselines leak mutual information between the model and the output, but I am not convinced. The adversary is label-only, and the setting targets edge devices. In such a setting, we typically protect the model weights from users because the model is trained by the service (app) provider and constitutes its intellectual property. Users may attempt to obtain the full model to avoid collecting training data or training a model themselves. Given this, how does the proposed FILOsofer protect the label? The user can always obtain the true label if they wish. Although the wrong label is outside the TEE, the true label inside the TEE must still be returned to the user; otherwise, the incorrect label would degrade service quality. Please further clarify the method and the results in Table 2. If the adversary uses the true labels for stealing, I do not believe the results would remain as low as 10%, 1%, or 0.5%.
> >
> > Regarding the responses to the listed weaknesses:
> >
> > 1. Concern addressed. Please include the results in the main paper, as GroupCover is significantly more secure than the baselines chosen in the current version.
> >
> > 2. Partially addressed. I agree this is a contribution, but it is minor. That more query budget leads to more severe model stealing is largely expected.
> >
> > 3. Not addressed. In GroupCover, 50 queries are used per class, meaning 500 queries for CIFAR-10 and 5000 for CIFAR-100. It is ok that this paper uses 50 and 5000 queries for the entire dataset. However, the paper claims that it “follows the same setting in Section 6 with both configuration and utility cost metric.” Section 6 contains multiple configurations for different baselines. Which configuration is actually used for the evaluation?
> >
> > 4. Not addressed. I still do not understand how LoRA recovers the true labels. Could you expand Equation 9 with more details? If the attacker knows which part corresponds to LoRA and uses initialized weights for those layers, the adversary would obtain the original earlier layers and only need to steal the remaining partial layers. Wouldn’t this make the leakage even more severe?
> >
> > 5. Not addressed. This issue aligns with my main concern above.

---

> > > ### Author Response · Authors · 2025-11-24
> > > **Comment about the main concern**
> > >
> > > Thank you for your comment and the opportunity to clarify. We believe the source of the misunderstanding lies in the distinction between authorized and unauthorized users. Our method provides black-box protection for authorized users while offering significantly stronger protection for unauthorized users by outputting a constant label. In addition, our framework supports a pay-per-query mechanism that can limit the number of model queries, ensuring long-term protection even under black-box access. Note that none of the prior TSDP-based methods (a) can distinguish between authorized and unauthorized users, and (b) can enforce query limits at the user level. This is the key difference between this work and prior art. There was no consideration given to the distinction between authorized and unauthorized actions in previous work, and we are the first to address this issue.
> > > Our threat model is analogous to a software licensing scenario (and in fact similar to commercial AI tools such as ChatGPT): a legitimate (authenticated) user who has paid for the license can make a predetermined number of queries—or unlimited queries under a full license—while unauthorized users or external adversaries cannot access the model without proper authentication. For smaller models (e.g., CNNs), a full-license model may be more appropriate, while for larger and more complex models (e.g., large CNNs and LLMs), a pay-per-query approach is more effective since model reconstruction (under black-box) would require a significantly large number of queries.
> > > To clarify:
> > > The results in Table 2 correspond to unauthenticated attackers, who can only observe obfuscated outputs.
> > >
> > >
> > > If an attacker were given access to true labels (authenticated queries), the protection would be similar to a black-box (hence lower), as authenticated access implies trust and license authorization.
> > >
> > >
> > > We will revise Section 3 and the caption of Table 2 to clearly distinguish between authenticated and unauthenticated queries.

---

> > > > ### Comment · Reviewer_cuad · 2025-11-26
> > > > **Response to authors**
> > > >
> > > > Thanks again for the author’s clarification. However, I believe it is unnecessary to distinguish whether the user is authorized. From the model owner’s perspective, the goal should be to protect privacy under strong adversaries. In particular, it is tricky and unfair to evaluate the proposed method on unauthorized users while evaluating the baselines on authorized users. An adversary can appear in any position in the system, and in real-world scenarios, authorized users are far more common.
> > > >
> > > > If we indeed assume that the adversary is only unauthorized, then several baselines would already provide the same level of protection. For instance, DarkneTZ places the last several layers inside the TEE, so an unauthorized adversary cannot obtain these layers or the outputs. Thus, without access to the complete model, the outputs are effectively random. Similarly, for GroupCover and NNSplitter, the outputs are protected inside the TEE, so the adversary also fails. But in Table 2, it appears that the baselines allow the attacker to obtain the true outputs, while the attacker cannot retrieve them when attacking FILOsofer. This makes the evaluation unfair and largely meaningless.

---

> ### Author Response · Authors · 2025-11-24
> **Comments regarding weaknesses**
>
> 1- Thank you! We will add these results.
>
> 2- We will tone down the claim regarding the novelty of this contribution. However, we maintain that the implications of our findings are significant, as they empirically demonstrate the inherent limitations of existing TSDP frameworks and underscore the need for a fundamentally different design approach. This insight provides an important motivation for our proposed solution.
>
> 3- All these papers (ours, TEESlice, and GroupCover) follow the original KnockoffNet setting and deploy the KnockoffNet codebase. Importantly, KnockoffNet uses an adaptive sampling strategy, meaning that the adversary repeatedly selects query inputs from the candidate pool using random sampling without any class-wise control. Consequently, this strategy does not guarantee that the final transfer set contains an equal number of samples for each class.
> I also examined the exact code used in GroupCover. In `knockoff/adversary/transfer.py` (lines 57–100, within the `get_transferset` function), at line 62 they also use:
> ```python
> idxs = np.random.choice(list(self.idx_set), replace=False, ...)
> ```
> Because we use a balanced dataset such as CIFAR-10, where each class originally contains the same number of images, the resulting class distribution in the transfer set will **tend to be approximately balanced in expectation**, although exact equality is not guaranteed.
>
> 4- Sorry for the confusion. We will modify it in the revised version. Equation (9) defines the final prediction during secure inference:
> $\hat{y} = f_{\mathrm{TEE}}(Z^{(\ell_s)}; A,B) + \tilde{y},$
> where: ($Z^{(\ell_s)}$) is the entry-layer activation at layer ($\ell_s$), computed by the obfuscated backbone inside the REE. ($\tilde{y} = f_{\mathrm{REE}}(X; W')$) is the preliminary, degraded prediction produced by the obfuscated weights ($W'$). ($(A,B)$) are the cross-layer LoRA parameters inside the TEE, representing a low-rank task-specific correction. The ($\tilde{y}$) is the wrong answer unauthorized users can obtain in the REE, but with the correction ($f_{\mathrm{TEE}}(Z^{(\ell_s)}; A,B) $) calculated in TEE, the ($\hat{y}$) can be accurate.
> We agree that the described scenario represents a much stronger adversary model, where the attacker has knowledge of our whole mechanism including the location of obfuscated layers and LoRA. Under such an extreme scenario, any defense would be difficult to maintain. We believe in this scenario, obfuscating more layers can help, so that recovery needs more data. And as the attacker cannot obtain the right input-output pairs, it still be very hard to recovery. For adaptive attackers, we evaluate two more practical scenarios in Section 7: 1) adapting norm clipping to the weight level, where the adversary constrains weight perturbations within a scaled range of the modified parameters. 2) we consider a Fisher‑aware adversary who knows that Fisher Information is employed for obfuscation but remains unaware of which specific layers are protected. In this case, the adversary computes Fisher Information across the entire model to approximate the defender’s masking strategy and mount a more globally informed attack. The results show that our method is robust to the adaptive attacks.
>
> 5- To clarify, the results presented correspond to unauthorized users, who only observe obfuscated outputs. The confusion likely arises from the distinction between true labels (available only through authorized access within the TEE) and obfuscated results (visible to unauthorized users). We agree that an authorized user effectively interacts with the system as a standard black-box model. We will revise the table caption and discussion to make this distinction explicit.

---

> ### Author Response · Authors · 2025-11-27
> **Comment about the main concern**
>
> We thank the reviewer for this thoughtful comment and the opportunity to clarify our evaluation setup. We evaluate our methods on the same query dataset with the authorized labels to assess its robustness. After running the new experiments with a scale factor of 0.1(C10, C100), 0.5(ImageNet200) and a weight ratio of 5e-4, for ViT-B/16, we applied a scale factor of 0.01 and a weight ratio of 1e-4, We find that our methods continue to provide protection against KnockoffNets, particularly when the number of queries is small. Even when the query count increases to 5,000, the performance remains strong. Compared with black-box, the obfuscated weights will be harder to recover.
>
> As shown in Figure 1, when the number of queries increases, an adversary—even without access to TEE-protected layers or labels—can still successfully construct a surrogate model by solely observing the information available outside the TEE (e.g., GPU-visible weights, GPU runtime, and embedding-domain leakage). In our evaluation of DarkneTZ, we did place the last three layers inside the TEE, consistent with the original design. The reported attack results correspond to an adversary who uses only publicly available information (not the protected labels or TEE-internal values) to reverse-engineer the protected layers. This demonstrates that protecting only a few layers—whether early (Serdab), late (DarkneTZ), or intermediate (TEESlice)—is insufficient once the adversary is allowed a large number of queries.
>
> Our approach addresses this fundamental limitation by adopting a holistic obfuscation strategy that removes exploitable information in the externally observable output. Table 1 further provides a systematic overview of all feasible obfuscation strategies, showing that we evaluated and compared the precise configurations the reviewer references.
>
> | Model | Dataset | Query 50 | Query 5000 |
> | :--- | :--- | :---: | :---: |
> | **VGG19** | C10 | 10.00 | 17.25 |
> | | C100 | 1.00 | 1.00 |
> | | ImageNet200 | 0.50 | 0.50 |
> | **Resnet18** | C10 | 10.00 | 18.51 |
> | | C100 | 1.00 | 1.00 |
> | | ImageNet200 | 0.50 | 0.50 |
> | **ViT-B/16** | C10 | 10.31 | 14.87 |
> | | C100 | 1.41 | 6.83 |
> | | ImageNet200 | 0.85 | 2.21 |

---

> > ### Comment · Reviewer_cuad · 2025-11-28
> > **Response to authors' comment**
> >
> > Thank you for providing the additional experiments on the authorized adversary. I now understand and appreciate the strong defense performance demonstrated by the proposed FILOsofer. These results directly address my concerns regarding its protection capability.
> >
> > However, regarding the second paragraph of the comment, I am still confused. In particular, I do not fully understand the conclusion: “This demonstrates that protecting only a few layers … is insufficient once the adversary is allowed a large number of queries.” The proposed FILOsofer also protects only a few layers (the last five layers, if I am not mistaken).
> >
> > Here is my current understanding; if anything is incorrect, please feel free to point it out:
> >
> > **If the adversary is only unauthorized (the main setting of the paper):**
> >
> >  They cannot get the weights and output label (if the method protect label) in TEE. As noted in Section 3, “The collected input–output pairs are then used to train the surrogate model.” However, under the unauthorized-adversary assumption, the attacker cannot obtain these input–output pairs. For **FILOsofer**, the attacker cannot observe the true output labels (because they are protected in the TEE). Thus, the attacker can only use the model parameters outside the TEE to directly estimate the MS accuracy. Since the model outputs incorrect labels, the MS accuracy naturally collapses to random (1 / num_classes). For **DarkneTZ**, the last three layers are placed inside the TEE. This means the visible model outside the TEE is incomplete, the missing layers is random initialized. Importantly, the attacker also cannot obtain the true input–output pairs. Therefore, the evaluation procedure should be identical to FILOsofer: the attacker evaluate MS accuracy using only the model outside the TEE, which again produces random outputs. The expected MS accuracy should thus match that of FILOsofer. The same reasoning applies to the **black-box setting**. Regardless of how many queries the attacker makes, the MS accuracy should remain at random level because the attacker never observes the true labels.
> >
> > **If the adversary is authorized (the same setting as in previous works)**, then they are allowed to access the inputs and the true output labels, including those produced inside the TEE. In this case, the attacker can obtain the full input–output pairs required to train a surrogate model. Therefore, when the adversary is authorized, increasing the number of queries directly provides more training data, enabling the surrogate model to better approximate the protected model. As a result, the MS accuracy naturally increases as the query budget grows.
> >
> > Since the experiment shows that the proposed FILOsofer maintains good protection under large numbers of queries, is there something wrong with my understanding? Why do Black-box and DarkneTZ achieve high MS accuracy as the number of queries increases in the main paper?

---

> ### Author Response · Authors · 2025-12-04
> **Comment about the main concern**
>
> We appreciate the reviewer’s rigorous analysis of the adversary models and the opportunity to clarify the experimental observations. The reviewer is correct that under a strictly unauthorized setting—where the adversary observes neither the TEE-protected weights nor the true labels—the model stealing (MS) accuracy should theoretically collapse to random guessing for the black-box setting. Our experiments are designed to investigate a novel dimension of model protection: the influence of output types. We explicitly distinguish between obfuscated labels and true labels, providing the first comprehensive analysis of how this dual-output mechanism affects adversary capability.
>
> The divergence in protection capability between FILOsofer and DarkneTZ under this authorized setting highlights the core contribution of our approach. While DarkneTZ protects the final layers inside the TEE, it leaves the preceding layers in the REE. Consequently, the REE component functions as a high-quality feature extractor. An authorized adversary, equipped with correct labels and these clean features, can easily train a surrogate model to approximate the missing layers, leading to high MS accuracy as the query budget increases.
>
> In contrast, FILOsofer does not merely hide the layers; it actively obfuscates the parameters residing in the REE. Guided by our theoretical derivation using Fisher Information, we mathematically perturb the most sensitive weights, ensuring that the REE outputs are not clean features but rather degraded representations that maximize information loss relative to the task. Even when an authorized adversary obtains correct labels, the optimization landscape for training a surrogate model based on these perturbed features becomes significantly more challenging even with a large number of queries.
>
> Ultimately, this comparison underscores the critical trade-off between security and latency. Theoretically, one could achieve perfect protection with DarkneTZ by placing all layers inside the TEE (a true black box), but this would incur prohibitive latency overheads due to TEE memory and computation constraints. FILOsofer identifies an optimal trade-off point: by leveraging cross-layer LoRA and Fisher-guided obfuscation, we achieve a security level comparable to a full-TEE solution while maintaining the low latency characteristic of split execution. This allows us to provide robust protection against both unauthorized and authorized adversaries with little performance penalty.

---

### Official Review · Reviewer_n7a2 · 2025-10-31

**Soundness:** 4
**Presentation:** 3
**Contribution:** 3
**Rating:** 8
**Confidence:** 3

**Summary:**

This paper proposes FILOSOFER, a TEE-shielded model partitioning framework that uses Fisher Information-guided weight obfuscation and cross-layer LoRA recovery to prevent model stealing on edge devices while keeping inference fast.

**Strengths:**

1) Tackles a real security gap in TEE-based model partitioning (information leakage under many queries).
2) Novel use of Fisher Information to select and perturb critical weights.
3) Cross-layer LoRA recovery offers strong utility with very low overhead.
4) Evaluation is comprehensive with multiple models, datasets, Jetson hardware, and adaptive attacks.
5) Practical and lightweight design suitable for real-world edge and LLM deployment.

**Weaknesses:**

1) Experiments lack recent  model-stealing baselines beyond KnockoffNet.
2) Analysis limited to query based attacks, side-channel attacks which are very prevalent should also be explored.

**Questions:**

1) The paper clearly identifies a key weakness in prior TSDP frameworks, which is residual information leakage under repeated queries. For this the authors propose an elegant, low-cost solution. The Fisher-guided obfuscation idea is clever and fits well with LoRA’s lightweight recovery, making the system both secure and fast.
2) However, the work needs stronger theoretical justification for why Fisher perturbation guarantees output uniformity and resistance to adaptive querying.
3) The evaluation is broad but mostly limited to one type of attacker; it would be more convincing to test stronger, query-adaptive or side-channel-aware methods as well.
4) Lastly, the LLM section feels preliminary. I feel expanding to larger models or analyzing memory–latency scaling would improve completeness.

---

> ### Author Response · Authors · 2025-11-21
> **Answering the questions**
>
> We thank the reviewer for the careful reading of our paper and for the constructive comments.
>
> 1. The paper clearly identifies a key weakness in prior TSDP frameworks, which is residual information leakage under repeated queries. For this the authors propose an elegant, low-cost solution. The Fisher-guided obfuscation idea is clever and fits well with LoRA’s lightweight recovery, making the system both secure and fast.
>
> We thank the reviewer for the positive feedback and for recognizing our key contributions.
>
> 2. However, the work needs stronger theoretical justification for why Fisher perturbation guarantees output uniformity and resistance to adaptive querying.
>
> We appreciate this insightful comment. We agree that a stronger theoretical justification of why Fisher-guided perturbation contributes to output uniformity and resistance to adaptive querying would improve the paper. Our intuition is that Fisher Information identifies parameters that have the greatest influence on the model’s output distribution, and perturbing these selectively minimizes the sensitivity of outputs to input variations, thereby reducing exploitable signal for adaptive attacks. We will expand the analysis provided in the Appendix (Section 9.10). We give the simplified version here: **Fisher information measures how sensitive the model output is to each weight. Perturbing weights along directions of high Fisher value affects the output most strongly**. Using a first-order Taylor expansion, the change in the target-class score is well approximated by the gradient: ($\Delta s \approx \nabla_W s(X;W)^\top \Delta W$). Therefore, a perturbation aligned with ($\nabla_W s$) and scaled by ($F^{-1}$) both increases the target score and reduces variance of outputs across inputs, approximating lower mutual information ($I(X;Z)$). This makes the output distribution more uniform and limits the information gain of adaptive queries, providing a principled, Fisher-guided obfuscation.
>
> 3. The evaluation is broad but mostly limited to one type of attacker; it would be more convincing to test stronger, query-adaptive or side-channel-aware methods as well.
>
> Thanks for your suggestion. We consider two adaptive attacker strategies in Section 7 and **demonstrate that our defense is effective to these informed adversaries**. The two stronger attackers are 1) adapting norm clipping to the weight level, where the adversary constrains weight perturbations within a scaled range of the modified parameters. 2) a Fisher‑aware adversary who knows that Fisher Information is employed for obfuscation but remains unaware of which specific layers are protected. In this case, the adversary computes Fisher Information across the entire model to approximate the defender’s masking strategy and mount a more globally informed attack.
>
> We also test the membership inference attack (MIA) in Appendix 9.4.
>
> We agree that side-channel attacks could be a possibility for an attacker; however, we assume that such a threat can be mitigated by utilizing orthogonal solutions such as physical shielding (to prevent physical side-channel attacks such as power or EM) and/or proper cache/memory isolation techniques to prevent side-channel leakage in TEEs. We will clarify this distinction in the revised manuscript and discuss potential directions for integrating side-channel resistance in future work.
>
> 4. Lastly, the LLM section feels preliminary. I feel expanding to larger models or analyzing memory–latency scaling would improve completeness.
>
> Thanks for your suggestions! Based on our assumptions and real-world scenario, we deployed the system in edge-devices, which have limited storage and capacity. We do a primary latency test on Jetson in Table 6. We also add new experiments on LLMs analyzing the sensitiveness of layers of our obfuscation which may make it more complete!

---

> > ### Comment · Reviewer_n7a2 · 2025-11-26
> >
> > We appreciate the clarifications. A formal formulation of the effect of optimizing Fisher information on the distribution will be interesting.

---

> > > ### Author Response · Authors · 2025-11-27
> > > **Formal Formulation of Fisher Information**
> > >
> > > We appreciate your comments and suggestions!
> > >
> > > 1.1 Notation and Setup
> > > Let $x \sim \mathcal{D}$ denote inputs from a data distribution, and consider a conditional probabilistic model $p(y \mid x; W)$. We analyze the effect of perturbing parameters $W$ to $W + \Delta W$, assuming $\|\Delta W\|$ is sufficiently small.
> > > We define the following quantities:
> > >
> > > Perturbed Output: $p_{W+\Delta W}(z \mid x)$ denotes the conditional output distribution after perturbation.
> > >
> > > Perturbed Marginal: $p_{W+\Delta W}(z) = \int p_{W+\Delta W}(z \mid x) p(x)  dx$.
> > >
> > > Score Function: $g_{L_t}(x) := \nabla_W \log p(L_t \mid x; W)$ is the gradient of the log-likelihood for a target class $L_t$.
> > >
> > > Target Score: $s(x; W) := \log p(L_t \mid x; W)$.
> > >
> > > 2. Theoretical Analysis
> > >
> > > 2.1 Information Theoretic Identities
> > >
> > > Lemma 1 (Mutual Information Identity).
> > >
> > > For any joint distribution $p(x, z)$, the mutual information can be expressed as the expected KL divergence between the conditional and marginal distributions:
> > >
> > > $I(X; Z) = E_{x \sim D} [ D_{KL}(p(z \mid x) || p(z)) ].$
> > >
> > > Lemma 2 (Local KL Expansion / Conditional Fisher).
> > > For a small perturbation $\Delta W$, the KL divergence admits the following quadratic expansion:
> > > $$D_{\mathrm{KL}}\big(p(\cdot \mid x; W) \,\|\, p(\cdot \mid x; W+\Delta W)\big) = \frac{1}{2} \Delta W^\top F(x; W) \Delta W + o(\|\Delta W\|^2),$$
> > > where $F(x; W)$ is the Conditional Fisher Information Matrix:
> > > $$F(x; W) := \mathbb{E}_{z \sim p(\cdot \mid x; W)} \Big[ \nabla_W \log p(z \mid x; W) \, \nabla_W \log p(z \mid x; W)^\top \Big].$$
> > > Proof.
> > > By performing a Taylor expansion of the log-likelihood and utilizing the property that the score function has zero mean, the first-order term vanishes. The leading term is quadratic in $\Delta W$. Standard derivations in asymptotic statistics yield the displayed form. $\square$
> > >
> > > 2.2 Main Theorem
> > >
> > > Theorem 1 (Local Quadratic Approximation of Mutual Information).
> > >
> > > For sufficiently small $\Delta W$, the mutual information of the perturbed model is given by:
> > >
> > > $I_{W+\Delta W}(X; Z) = I_W(X; Z) + \frac{1}{2} \mathbb{E}_{x \sim \mathcal{D}} \big[ \Delta W^\top F(x; W) \Delta W \big] + o(\|\Delta W\|^2).$
> > >
> > > Proof.
> > > Using Lemma 1, $I_{W+\Delta W}(X; Z) = E_x [D_{KL}(p(\cdot \mid x; W+\Delta W) \| p_{W+\Delta W}(\cdot))]$. We expand the integrand around $W$, accounting for variations in both the conditional $p(\cdot \mid x; W+\Delta W)$ and the marginal $p_{W+\Delta W}(\cdot)$ with respect to $\Delta W$. Through careful bookkeeping of first- and second-order terms, and applying Lemma 2 for the conditional contribution, the stated quadratic term emerges as the dominant second-order contribution. The remainder is $o(\|\Delta W\|^2)$. $\square$
> > >
> > > 2.3 Optimal Perturbation
> > >
> > > Proposition 1 (Optimal Local Perturbation under Fisher Budget).
> > > Define the population-averaged conditional Fisher matrix as $\overline{F} := \mathbb{E}_{x \sim \mathcal{D}}[F(x; W)]$. Consider the following constrained optimization problem (quadratic-budget approximation):
> > >
> > > $\max_{\Delta W} E_x [g_{L_t}(x)]^\top \Delta W ~~~~~\text{s.t.}  \Delta W^\top \overline{F}\Delta W \le \varepsilon$
> > >
> > > If $\overline{F}$ is positive definite, the optimal perturbation direction is:
> > > $\Delta W^\star \propto \overline{F}^{-1} \, E_x[g_{L_t}(x)].$
> > >
> > > Proof.
> > > This formulation represents a linear objective maximized under a quadratic constraint. The Lagrangian is given by:
> > >
> > > $L(\Delta W, \lambda) = E_x[g_{L_t}(x)]^\top \Delta W - \lambda (\Delta W^\top \overline{F} \Delta W - \varepsilon).$
> > >
> > > The stationarity condition $\nabla_{\Delta W}\mathcal{L} = 0$ yields $E_x [g_{L_t}(x)] = 2\lambda \overline{F} \Delta W$. Assuming $\lambda > 0$ and $\overline{F}$ is invertible, the result follows directly. $\square$
> > >
> > > 2.4 Remarks on Theoretical Results
> > >
> > > Metric Consistency: The matrix $\overline{F}$ provides the theoretically correct second-order (KL) metric for quantifying distributional change induced by $\Delta W$. Using an uncentered class-specific matrix (e.g., $F_{L_t}^{(\text{raw})} = E[g_{L_t}g_{L_t}^\top]$) is generally inconsistent with the KL expansion without additional strong assumptions.
> > > Variance Targeting: If the objective is to directly target the sample-wise variance of the target score, the appropriate quadratic cost is $\Delta W^\top F_{L_t} \Delta W$, where $F_{L_t}$ is the centered class-covariance matrix.

---

> > > ### Author Response · Authors · 2025-11-27
> > > **Formal Formulation of Fisher Information**
> > >
> > > 3. Practical Implementation and Heuristics
> > >
> > > 3.1 Feasibility
> > >
> > > Finite Dataset: For a finite dataset $\{x_i\}_{i=1}^m$, strictly enforcing $s(x_i; W+\Delta W) = c$ reduces to a linear system $G \Delta W = b$. If the parameter dimension is large, a minimum-norm solution usually exists.
> > >
> > > Independence: Achieving exact independence $p(z \mid x) = p(z)$ is generally impossible with finite-dimensional parameters. Thus, we minimize distributional proxies (e.g., variance, mutual information, empirical KL).
> > >
> > > Approximation: In practice, $\overline{F}$ and $\mathbb{E}[g]$ are estimated empirically, and $F^{-1}$ is approximated to reduce computational cost.
> > >
> > > 3.2 Targeted Fisher for Obfuscation
> > >
> > > To steer the model toward a target label $L_t$ while reducing input dependence, we define a gradient-based sensitivity measure:
> > > $F_{L_t} = \mathbb{E}\left[ (\frac{\partial \mathcal{L}(x, W)}{\partial W})^2  |  y=L_t \right].$
> > > Note: This is a non-standard, heuristic Fisher matrix. It captures weights that most strongly influence the output toward $L_t$.
> > >
> > > Perturbation via Gradient Update
> > >
> > > Instead of the computationally expensive natural gradient ($\overline{F}^{-1}\nabla$), we apply a perturbation along the standard gradient of the target loss:
> > >
> > > $W \leftarrow W + \eta \cdot \nabla_W \mathcal{L}(x, W),$ where $\eta$ is a scale factor.
> > >
> > > This strategy approximates mutual information minimization through the following mechanism:
> > > Fisher-guided Selection: Weights with high $F_{L_t}$ are the most sensitive control knobs for the output.
> > > Gradient Alignment: The perturbation maximizes the target score increase.
> > > Variance Reduction: By forcing the model to confidently predict $L_t$ for all inputs (thereby increasing the target score globally), the variance of the output distribution across inputs $x$ decreases.

---

> ### Author Response · Authors · 2025-11-21
> **Addressing the Weaknesses**
>
> We thank the reviewer for this comment~ Below we address the main concern.
>
> Weakness 1: Thank you for this comment. Our model-stealing evaluation follows prior work[1][2], which consistently adopts the KnockoffNet attack as a representative and well-established benchmark. Using the same attack enables a fair and direct comparison with existing literature. To further strengthen our evaluation, we have also included results for membership inference attacks (see Section 9.4 in the Appendix), demonstrating that our proposed defense is effective across multiple attack types.
>
> Weakness 2: We appreciate the reviewer’s observation. Our analysis primarily focuses on query-based model extraction attacks, which represent the most direct threat to TEE-shielded inference systems. We agree that side-channel attacks (e.g., power, EM, or cache-based leakage) are an important class of threats. However, these attacks exploit physical or microarchitectural channels and are largely orthogonal to the query-based leakage problem we study. Such threats can be mitigated using existing hardware-level countermeasures, including physical shielding, noise injection, and cache/memory isolation in TEEs. We will clarify this scope distinction in the revision and outline potential directions for extending our framework to incorporate side-channel–aware protection mechanisms in future work.
>
>
> [1]. Zhang, Z., Gong, C., Cai, Y., Yuan, Y., Liu, B., Li, D., ... & Chen, X. (2024, May). No privacy left outside: On the (in-) security of tee-shielded dnn partition for on-device ml. In 2024 IEEE Symposium on Security and Privacy (SP) (pp. 3327-3345). IEEE.
> [2]. Zhang, Z., Wang, N., Zhang, Z., Zhang, Y., Zhang, T., Liu, J., & Wu, Y. (2024, July). Groupcover: a secure, efficient and scalable inference framework for on-device model protection based on tees. In Forty-first international conference on machine learning.

---

### Official Review · Reviewer_9xvv · 2025-11-02

**Soundness:** 3
**Presentation:** 3
**Contribution:** 3
**Rating:** 6
**Confidence:** 4

**Summary:**

This paper presents FILOsofer, a framework for protecting deep neural networks deployed on edge devices from model stealing attacks. The authors demonstrate that existing TSDP methods remain vulnerable when attackers have large query budgets, as they gradually leak information through accurate outputs. FILOsofer addresses this by using Fisher Information to selectively change critical weights, forcing the model to produce uniform outputs, while a lightweight cross-layer LoRA module stored in the TEE restores the model performance to authorized users. Experimental results show FILOsofer achieves 10x better security against model stealing with 50x lower computational overhead compared to prior TSDP solutions.

**Strengths:**

1. Exciting application of TEEs in protecting DNNs
2. Very strong motivation of the work, and good presentation of background (some exceptions mentioned below)
3. Comparison with SOTA related approaches
4. Both theoretical and practical execution, with on-device experiments on a ARM based Jetson machine

**Weaknesses:**

1. Your thread model assumes that the adversary can infer the models architecture by monitoring the weights in REE space. Is that reasonable? Related works that you mention protect one or even more layers inside the TEE. Are you considering a model without the protected layers? If not, how can you know the architecture of the protected layers (length, and number of protected layers at the minimum).
2. While as mentioned above, the motivation is clear and background section provides good info to the reader, it took me a while to understand the FILOsofer aims to protect the model parameters from an attacker. I suggest you have a quick reference of your thread model earlier in the manuscript. Same for L061 when you mention that the model. on GPU remains highly accurate, unless you know the related work it is not clear why this is important for you.
3. While I appreciate the implications section on applying FILOsofer on LLMs, I feel it was a rushed evaluation that comes out of the blue in the manuscript. There is no information about how they used the LLM for a classification task (and what exactly the task was). What was the reason that layer 15 was chosen, was it the most informative? reported the best accuracy? Have you tried other layers? What type of data have you used and what were the exact system prompts?

**Questions:**

1. Can you please clarify how an adversary can infer the architecture of the TEE protected layers?
2. Can you provide additional details about the application of FILOsofer on LLMs, as per my W3 comment?

---

> ### Author Response · Authors · 2025-11-21
> **Answering the questions**
>
> We thank the reviewer for the careful reading of our paper and for the constructive comments. Below we address the main questions.
>
> 1. Can you please clarify how an adversary can infer the architecture of the TEE protected layers?
>
> Thanks for your questions~ We assume that the adversary has **public-domain knowledge of the overall model architecture** (e.g.,ResNet18, ViT-B/16), including the structure of layers that are protected within the TEE. Previous work also studied how to infer the model architecture even with black-box settings[1]. This is a conservative assumption consistent with standard practices in security research, where the attacker is assumed to know the system design but not the secret parameters (eliminating the security by obscurity pitfall).
> Furthermore, as stated in Section 7, **we consider an even stronger adversary who is aware of the existence of the LoRA branch within our design**. These assumptions make our threat model intentionally stringent—if the defense remains effective even when the attacker has full architectural knowledge, it shows stronger robustness. In real-world deployments, where such detailed information is typically unavailable, our proposed defense would be even more secure.
>
> 2. Can you provide additional details about the application of FILOsofer on LLMs, as per my W3 comment?
>
> FILOsofer enables a **pay-per-query based framework on the edge device**. The user can establish credits (e.g., by paying money to the service provider). This credit is securely communicated to the TEE using establishing secure remote computation methods. Using this, TEE controls the output and only provides meaningful results if there is valid/unexpried credit.
>
> To achieve this, FILOsofer is adapted to label-based output, so for LLMs, we first use AutoModelForSequenceClassification to add one classifier layer. However, we find that directly applying pretrained LLMs to datasets like Arc_easy, SciQ, the accuracy is not high. So we firstly fine-tuned the LLMs only on the last classifier layer. But for Arc_easy, the accuracy is still quite low, so we finally do 3-epoch sft on the whole model and gain 99% accuracy on this dataset. For each dataset, we make the system prompts as: “Question:”+dataset[“question”]+datasets[“options”]. Then we choose the last layer, which is layer 15 and applies obfuscation on this layer and the classifier layer to lower the accuracy and applies the LoRA branch to recovery.
> We also add some new experiments  on LLMs to test the influence of different layers.
>
> [1]. Oh, S. J., Augustin, M., Fritz, M., & Schiele, B. (2018, February). Towards Reverse-Engineering Black-Box Neural Networks. In International Conference on Learning Representations.

---

> ### Author Response · Authors · 2025-11-21
> **Addressing the Weaknesses**
>
> We thank the reviewer for this comment~ Following are the answers.
>
> Weakness 1: As noted in question 1, we adopt a conservative assumption that favors the adversary in order to evaluate our defense **under the most challenging conditions**. To further support this point, we will include an ablation study demonstrating the effect of varying the attacker’s architectural knowledge on the effectiveness of our defense.
>
> Weakness 2: We appreciate this helpful suggestion. We will revise the introduction to include a brief summary of our threat model early in the manuscript to make it clear that **FILOsofer aims to protect model parameters from an adversary attempting model extraction or reconstruction**. We will also clarify the sentence in Line 061. Our intent was to convey that, even when existing obfuscation methods are applied, a surrogate model can still improve its reconstruction accuracy as the number of queries increases, due to continued information leakage from model outputs. The term “accurate” was poorly chosen—we will rephrase it to emphasize information leakage and query-based improvement, rather than suggesting high baseline model accuracy.
>
> Weakness 3: Details are provided in question 2 and we also add new experiments on the sensitiveness of layers!

---

### Official Review · Reviewer_nNxR · 2025-11-03

**Soundness:** 2
**Presentation:** 2
**Contribution:** 2
**Rating:** 2
**Confidence:** 4

**Summary:**

The paper introduces FILOsofer, a defense against model stealing in on-device deployments where a user (as an atatcker) can query the target model deployed on their device, see predictions and model weights. The proposed defence partitions the model between a Trusted Execution Environment (TEE) and the Rich Execution Environment (REE):
- A cross-layer LoRA kept and executed inside the TEE protecting confidentiality of model weights
- The Fisher-guided obfuscated weights kept in the REE.

The two components are trained with a constraint-aware joint objective to balance security and utility: ensuring the obfuscated backbone resists trivial recovery while the TEE-resident LoRA restores accuracy for authorized use.

**Strengths:**

Empirically analyzing vulnerabilities of TEE-Shielded DNN Partitioning to model stealing attacks.

**Weaknesses:**

1. Section 3 appears to combine two sources of attacker leverage: (i) the model’s architecture/weights in REE so the adversary first infers the protected model’s architecture and weights from publicly available models, and (ii) carefully chosen queries to collect outputs so the attacker issues limited queries on carefully selected inputs and records the corresponding outputs. The proposed backbone obfuscation does not help with (ii). Line 179-1180 discusses results of existing methods as follows:the partitioned model executed on GPUs remains accurate, enabling attackers to initialize surrogate models effectively. However, no evidences of that exist in Figure 1 which only studies the impact of number of queries.

2. The obfuscation optimisation aims to make the obfuscated model’s output to be input-independent.

3. The proposed defences requires joint-training without any discussion on data for this training.


4. TEE-Shielded DNN Partitioning has been extensively studies in the literature.

5. This paper mixes privacy of training data and IP/confidentiality of the model. DP and MPC do not address this problem of model stealing, statements in intro are problematic such as  ``To mitigate these security risks, researchers have explored two defense strategies: (i) Cryptographic approaches: Methods such as Multi-Party Computation (MPC) (Juvekar et al., 2018), Homomorphic Encryption (HE) (Gilad-Bachrach et al., 2016; Kim et al., 2022), and Differential Privacy (DP) (Abadi et al., 2016; Girgis et al., 2021) aim to safeguard both input data and model parameters through algorithmic guarantees. D''

6. The paper is not motivated very well: ``but executing entire models within TEEs is inefficient and slow'' --> It is not really the case anymore given recent GPUs with TEE supports

7. line 058: confidentiality(Zhou et al., 2023; Sun et al., 2024). --> typo

**Questions:**

1. Is the model stealing due to output or model weights?

2. If the goal is to make the obfuscated model’s output to be input-independent, why not just using randomly inisialised weights? why do you need Fisher Information?

3. Which datasets you need for the joint-training algorithm? Do you need the whole training dataset to protect against model stealing? If so, how practical and costly it is?

 4. Which layers are most sensitive layers?

5. How robust it is to an informed adversary?

---

> ### Author Response · Authors · 2025-11-21
> **Answering the questions**
>
> We are very thankful for your comments and questions~ Below we address the main concerns.
>
> 1. Is the model stealing due to output or model weights?
>
> The goal of the attacker in our setting is to **steal the model weights and reconstruct the model**. Our key insight is that the number of queries made by the adversary significantly impacts the accuracy of model reconstruction, as illustrated in Figure 1. This occurs because the model’s outputs inherently leak information about its internal parameters, enabling an attacker to train a surrogate model that closely approximates the original. To prevent this leakage, our defense ensures that the model’s output remains constant and uninformative for unauthenticated users, and only produces meaningful outputs when the user is verified via the TEE.
>
> 2. If the goal is to make the obfuscated model’s output to be input-independent, why not just using randomly inisialised weights? why do you need Fisher Information?
>
> Our goal is to **balance the utility–privacy–latency tradeoff rather than simply make the model’s output input-independent**. The LoRA branch is responsible for recovering the original output once the user is authenticated. If too many weights are obfuscated, maintaining utility requires a larger (higher-rank) LoRA branch, which increases recovery latency—as shown in Table 3. To minimize this tradeoff, we seek to obfuscate as few weights as possible while still ensuring privacy. Fisher Information provides a principled way to identify the most influential parameters for obfuscation, achieving stronger privacy with fewer modified weights compared to random selection. These effects are also further studied in the Appendix, Sections 9.5 and 9.6. We will also include an ablation study comparing Fisher-based and random obfuscation to quantify their impact on utility, privacy, and latency.
>
> 3. Which datasets you need for the joint-training algorithm? Do you need the whole training dataset to protect against model stealing? If so, how practical and costly it is?
>
> Joint training uses the same setup as the LoRA-branch fine-tuning. For the obfuscation component, **we only perform a single forward pass to calculate the Fisher information of each weight, which introduces negligible overhead**. In contrast, the LoRA branch requires fine-tuning and therefore needs access to the corresponding training dataset (e.g., CIFAR-100). We use the whole training datasets for fine-tuning to gain highest accuracy. To maintain efficiency while preserving effectiveness, rather than attaching a separate LoRA module to every layer, we design a cross-layer LoRA branch that spans all obfuscated layers, significantly reducing both parameters and training cost.
> Compared with prior obfuscation methods such as TEESlides and NNSplitter, **our computational cost is substantially lower and the overall process is more stable**. TEESlides requires iterative slice pruning and repeatedly training the pruned model, while NNSplitter relies on reinforcement learning to identify layers and weights, often requiring many search rounds. In contrast, our approach needs only one Fisher pass plus lightweight LoRA finetuning, making it significantly more efficient.
>
> 4. Which layers are most sensitive layers?
>
> Our experiments in Appendix 9.8 show that **the set of sensitive layers and weights varies across datasets and target labels, and the sensitive layers can be identified by fisher information**. The fisher information of the sensitive part weights are much larger than the others, which suggest by adding perturbations on these weights can largely affect the output. In FILOsofer, the last layer must be perturbed because it directly determines the logits and model outputs. To keep the efficiency, we use a cross-layer LoRA branch shared across several layers; Therefore, to jointly ensure output protection and accurate cross-layer LoRA recovery, we perturb specifically the last few layers rather than simply the top-sensitivity ones.
>
> 5. How robust is it to an informed adversary?
>
> We study two new attack scenarios for an informed/adaptive adversary in Section 7 and **demonstrate that our defense is effective to these informed adversaries**. 1) adapting norm clipping to the weight level, where the adversary constrains weight perturbations within a scaled range of the modified parameters. 2) we consider a Fisher‑aware adversary who knows that Fisher Information is employed for obfuscation but remains unaware of which specific layers are protected. In this case, the adversary computes Fisher Information across the entire model to approximate the defender’s masking strategy and mount a more globally informed attack.

---

> ### Author Response · Authors · 2025-11-21
> **Addressing the weaknesses**
>
> 1: We appreciate the reviewer’s observation. To clarify, in our attack model, **the adversary has full knowledge of the model weights and architecture executed in the REE**. However, the attacker cannot access or observe the weights and LoRA branches residing inside the TEE. The adversary can issue unlimited queries with arbitrary inputs—there are no restrictions on query count or input selection. The key strength of our design lies precisely in providing robust protection against unlimited-query adversaries.
>
> 2: We appreciate the reviewer’s feedback. We will revise this sentence for clarity. Our intent was to convey that, **even when existing obfuscation methods are applied, a surrogate model can still improve its reconstruction accuracy as the number of queries increases**. This occurs because the model’s outputs continue to leak information about its internal weights. The term “accurate” was poorly chosen—we will rephrase it to explicitly emphasize the information leakage and query-based improvement rather than implying high baseline model accuracy.
>
> 3: Discussed in question 3.
>
> 4: Although these methods differ in how they partition and protect models, our analysis shows that **all existing TSDP approaches remain vulnerable to adversaries with large query budgets**, as demonstrated in Figure 1. We have extended our evaluation to include the most recent work in this domain, GroupCover, and confirmed that it exhibits the same vulnerability (see table in Comment). This consistent trend across both earlier and latest techniques underscores the necessity of a new design, such as the one proposed in our work, that can maintain robustness even under unlimited-query adversarial settings.
>
> 5: We appreciate the reviewer’s observation and agree that our introductory paragraph conflated training-data privacy with model confidentiality. We will revise this section for clarity. Specifically, we will remove the mention of DP, which focuses on data privacy, and expand our discussion of FHE and MPC. These cryptographic methods protect model confidentiality by enabling secure function evaluation, where both the input data and the model weights remain encrypted during computation (e.g., see [arXiv:2408.03561]). The revised text will clearly distinguish between data-centric and model-centric protection mechanisms.
>
>
> 6: We thank the reviewer for this comment. While we acknowledge recent advances in GPU-based TEEs, our work considers **a stronger adversary model in which the GPU kernel and runtime are untrusted**. Current GPU TEE solutions primarily protect data and computations at the memory or enclave boundary but do not provide isolation guarantees for the GPU kernel itself. To the best of our knowledge, neither prior research nor existing commercial systems address this level of threat. Therefore, our motivation remains valid, as our design aims to ensure protection even under this stronger, kernel-compromised adversarial setting.
>
> Weakness 7: Thanks for the feedback. We will fix that.

---

### Author Response · Authors · 2025-11-21
**New Experiments**

We thank for all reviewer for their constructive comments. The following are the experiments we added to make the paper more complete.

1. **The results of GroupCover[1]**. We add the attack results on GroupCover as shown in the table. We can observe that under Budget 5000, the attack is more effective, reaching XXX. For ResNet18 on C10, the attacker can recover 57.6% accuracy. But under small budget of 50, GroupCover achieves a good protection which is aligned to its paper. The inferior performance of GroupCover is because **GroupCover does not explicitly address the mutual information leakage between the model and its outputs, so its protection is not guaranteed in a principled way**. Consequently, while it performs well under small budget, the security it provides is not consistently ensured.


| Model    | Dataset     | Budget 50 | Budget 5000 |
| -------- | ----------- | --------- | ----------- |
| AlexNet  | C10         | 13.6      | 47.7        |
|          | C100        | 1.0       | 1.0         |
|          | ImageNet200 | 0.5       | 0.5         |
| ResNet18 | C10         | 10.0      | 57.6        |
|          | C100        | 1.0       | 19.7        |
|          | ImageNet200 | 0.5       | 6.3         |
| VGG19    | C10         | 10.0      | 11.7        |
|          | C100        | 1.0       | 1.3         |
|          | ImageNet200 | 0.5       | 2.9         |
| ViT-16   | C10         | 12.6      | 39.7        |
|          | C100        | 1.6       | 11.5        |
|          | ImageNet200 | 0.8       | 4.3         |


2. **Obfuscating individual layers in LLMs**. Keeping all other layers unchanged, we perturb only one layer at a time and measure the resulting model accuracy. Experiments are conducted on the SCIQ dataset, where the original accuracy is 0.92. We use a scale factor of 0.1 and a modified weight ratio of 1e^{-4}. **Our perturbation analysis shows that the early layers exhibit robustness to noise, whereas the middle and later layers are significantly more sensitive to perturbations.**

| Layer | Accuracy |
|-------|-----------|
| 0     | 0.885     |
| 1     | 0.593     |
| 2     | 0.452     |
| 3     | 0.324     |
| 4     | 0.434     |
| 5     | 0.443     |
| 6     | 0.281     |
| 7     | 0.359     |
| 8     | 0.213     |
| 9     | 0.194     |
| 10    | 0.248     |
| 11    | 0.350     |
| 12    | 0.246     |
| 13    | 0.306     |
| 14    | 0.241     |
| 15    | 0.286     |



3. **Sensitiveness between attention layers and MLP layers**. We use the same settings with scale factor of 0.1 and a modified weight ratio of 1e^{-4}. The following table shows the accuracy under attack, and we find **mlp layers are more sensitive to this gradient-based perturbation**.

| layer | attention | mlp  |
|-------|-----------|------|
| 0     | 0.889     | 0.239 |
| 1     | 0.540     | 0.305 |
| 2     | 0.485     | 0.254 |
| 3     | 0.329     | 0.299 |
| 4     | 0.461     | 0.245 |
| 5     | 0.307     | 0.254 |
| 6     | 0.332     | 0.286 |
| 7     | 0.321     | 0.240 |
| 8     | 0.291     | 0.239 |
| 9     | 0.308     | 0.273 |
| 10    | 0.258     | 0.239 |
| 11    | 0.305     | 0.239 |
| 12    | 0.239     | 0.239 |
| 13    | 0.317     | 0.239 |
| 14    | 0.239     | 0.239 |
| 15    | 0.289     | 0.239 |




[1]. Zhang, Z., Wang, N., Zhang, Z., Zhang, Y., Zhang, T., Liu, J., & Wu, Y. (2024, July). Groupcover: a secure, efficient and scalable inference framework for on-device model protection based on tees. In Forty-first international conference on machine learning.

---

### Meta-Review · Area_Chair_QvJF · 2026-01-11

**Summary:**

The paper proposed FILOsofer, a TEE-shielded DNN partitioning framework to protect on-device models against stealing attacks. The assumption is that most of the model weights are not inside the TEE. This method obfuscates Fisher-important weights in the exposed partition. The external model yields near-uninformative outputs, while a cross-layer LoRA module inside the TEE restores accuracy for authorized inference with low overhead. The results indicated improved extraction resistance and lower overhead than baselines. There are two clear positive reviews and two strong negative reviews. One of the negative reviewers engaged in the discussion and keep clear negative comments. After reading the comments, the AC found some key issues in the negative reviews which are not well addressed.

**Reviewer Concerns:**

Some main issues are summarized as follows.
- **Threat model/evaluation ambiguity (authorized vs unauthorized attackers).** (reviewer cuad and nNxR) It is unclear whether the claimed security holds under the strong/realistic adversary who gets correct labels/outputs. Reviewer cuad concerns authorized/unauthorized users. Reviewer nNxR also concern about what exactly causes stealing and what obfuscation actually blocks. The AC thinks this makes an unclear threat model since the security shifts (unauthorized-only v.s. authorized).
- **Method not convincingly justified.** (reviewer nNxR and n7a2) Why Fisher-guided perturbation is necessary vs simpler obfuscation and what guarantees output uniformity.
- **Security robustness scope incomplete.** (reviewer n7a2 and nNxR) Side-channel threat not evaluated; adaptive attacker coverage limited.

**Reviewer Scores:**

The AC believes that some of the questions are well addressed. But after reading answers, several key concerns (see summary above) are not well addressed and some discussions even make the threat models more unclear. The AC thinks the two clear negative reviewers are very likely to keep their score.

---

### Decision · Program_Chairs · 2026-01-26

Reject